# Ultrastrong exciton-plasmon couplings in WS$_2$ multilayers synthesized with a random multi-singular metasurface at room temperature

Tingting Wu[1,9], Chongwu Wang [1,9], Guangwei Hu [1,9], Zhixun Wang [1], Jiaxin Zhao[2], Zhe Wang[1], Ksenia Chaykun[2], Lin Liu[1], Mengxiao Chen [3], Dong Li [4], Song Zhu[1], Qihua Xiong [5], Zexiang Shen [2], Huajian Gao [4], Francisco J. Garcia-Vidal [6,7] ✉, Lei Wei [1] ✉, Qi Jie Wang [1,2] ✉ & Yu Luo [8] ✉

Van der Waals semiconductors exemplified by two-dimensional transition-metal dichalcogenides have promised next-generation atomically thin optoelectronics. Boosting their interaction with light is vital for practical applications, especially in the quantum regime where ultrastrong coupling is highly demanded but not yet realized. Here we report ultrastrong exciton-plasmon coupling at room temperature in tungsten disulfide (WS$_2$) layers loaded with a random multi-singular plasmonic metasurface deposited on a flexible polymer substrate. Different from seeking perfect metals or high-quality resonators, we create a unique type of metasurface with a dense array of singularities that can support nanometre-sized plasmonic hotspots to which several WS$_2$ excitons coherently interact. The associated normalized coupling strength is 0.12 for monolayer WS$_2$ and can be up to 0.164 for quadrilayers, showcasing the ultrastrong exciton-plasmon coupling that is important for practical optoelectronic devices based on low-dimensional semiconductors.

Exciton-polaritons owing to strong collective excitations of waves and electron-hole pairs, i.e., excitons have facilitated versatile applications in nonlinear optics, quantum photonics, condensed matter physics, and others. Various excitonic systems have been explored, including bulk III-V semiconductors and organic materials, which however either require cryogenic temperatures[1,2] to avoid the ionization of excitons at high temperatures or are susceptible to bleaching effects[3]. Recently,

van der Waals transition-metal dichalcogenides (TMDs) have emerged as new candidates for stable and atomically thin excitonic systems, offering numerous advantages, including direct bandgaps at visible frequencies, large exciton binding energies, pronounced resonance strengths, narrow linewidths even beyond room temperature[4,5], and strongly reduced structural disorder. In addition, strong coupling (SC) between excitons in TMD monolayers and photonic resonances has

[1]School of Electrical and Electronic Engineering, Nanyang Technological University, Singapore, Singapore. [2]School of Physical and Mathematical Sciences, Nanyang Technological University, Singapore, Singapore. [3]Zhejiang Provincial Key Laboratory of Cardio-Cerebral Vascular Detection Technology and Medicinal Effectiveness Appraisal, Zhejiang University, Hangzhou, China. [4]School of Mechanical and Aerospace Engineering, Nanyang Technological University, Singapore, Singapore. [5]State Key Laboratory of Low-Dimensional Quantum Physics and Department of Physics, Tsinghua University, Beijing, China. [6]Departamento de Física Teorica de la Materia Condensada and Condensed Matter Physics Center (IFIMAC), Universidad Autónoma de Madrid, 28049 Madrid, Spain. [7]Institute of High Performance Computing, Agency for Science, Technology and Research (A*STAR), Connexis 138632, Singapore. [8]National Key Laboratory of Microwave Photonics, Nanjing University of Aeronautics and Astronautics, Nanjing, China. [9]These authors contributed equally: Tingting Wu, Chongwu Wang, Guangwei Hu. ✉e-mail: fj.garcia@uam.es; wei.lei@ntu.edu.sg; qjwang@ntu.edu.sg; yu.luo@nuaa.edu.cn

been realized in all-dielectric microcavities with Fabry-Perot resonators[6,7], metal-based microcavities with reduced mode volumes[8,9], and plasmonic structures towards further enhancement of wave-matter couplings[10,11].

In a parallel scientific endeavour, ultrastrong coupling (USC) of light-matter interactions has been recently explored in various systems. Here, the normalized coupling strength, defined as $\eta = g/\omega_{ex}$ where $g$ is coupling strength and $\omega_{ex}$ is bare excitation energy, should be >0.1. Compared to SC, USC is a new regime of quantum light-matter interactions with faster control/response even at shorter device lifetimes and important for several quantum optoelectronic applications. However, since its early proposal[12,13], USC has been only observed in semiconductor quantum wells[14], superconducting circuits[15], Landau polaritons[16,17], organic molecules[18], phonons[19], and plasmons[20], most of which rely on coupling many dipoles to a cavity mode at cryogenic temperatures due to the technical challenges in implementation. Realizing USC under ambient conditions remains a challenge. Meanwhile, USC in monolayer TMDs has never been reached, which impedes their promise for ultrathin quantum devices[21]. This is because of the weak nanoscale matter excitations and the lack of significant transition dipole moments (typically in a few tens of Debyes) at such atomic thicknesses.

Here, we report the first observation of USC between excitons in a TMD monolayer and surface plasmons in a random multi-singular flexible plasmonic metasurface at room temperature. The dense singular nanometre-sized gaps are created by the cold etching technique and support strong field concentration. Multiple WS$_2$ excitons can interact with these deep-nanoscale plasmonic hotspots, thus promoting ultrastrong exciton-plasmon couplings. We observe a normalized coupling strength of 0.082 in unstrained TMD monolayers, which, via strain engineering in the flexible substrate, can be further tuned over a wide range from 0.075 to 0.12, then entering the USC regime. Furthermore, by increasing the number of WS$_2$ layers, the exciton-plasmon interaction shows increased normalized coupling strengths to 0.147 (trilayer) and 0.164 (quadrilayer) under strain. We believe that our reported strategy towards room-temperature USC in TMD multilayers could find applications in lasing, nonlinear optics, wearable optoelectronics, and other technologies on atomically thin platforms.

## Results

### Mechanism of USC in WS2 multilayers loaded with multi-singular metasurface

We start by discussing the fundamental ways to boost light-matter interaction. In general, in the case of collective ensembles, the light-matter coupling strength $g$ is related to the emitter number ($g \propto \sqrt{N}$) and the highly confined cavity field in a very small mode volume ($g \propto 1/\sqrt{V}$)[22,23]. Progress towards room-temperature SC with monolayer TMDs in dielectric-based microcavities has been limited by the inevitable increase in the emitter scattering rate ($\gamma \propto k_B T$) and by difficulties in reducing the cavity mode volume in dielectric structures due to the diffraction limit. To improve mode confinement with smaller mode volumes, surface plasmons have been used[23,24]. Hence, the pronounced room-temperature plasmon-exciton coupling is observed in periodic metallic structures supporting high-quality surface lattice resonances[25,26] with collectively excited large numbers of TMD excitons or in plasmonic resonators (such as dimers, gaps, etc.) with strong field enhancement in a small mode volume[27,28]. However, plasmonic lattices suffer from field delocalization, while localized resonances in small gaps or sharp nanostructures, which are usually sparse (the corresponding number of interacting emitters is reduced), pose a challenge for fine fabrication and have a limited number of plasmonic hotspots. To date, exciton-plasmon coupling strength with monolayer TMDs is typically in the range of 30–60 meV[29–31], far beyond

the regime of USC, where the coupling strength should be >0.1$\omega_{ex}$ (~200 meV).

Here we construct a cold-etched random multi-singular plasmonic metasurface grown on a flexible thermoplastic polyurethane (TPU) substrate (inset, left, Fig. 1a) as a platform to trigger USC coupling between excitons in two-dimensional (2D) TMDs and surface plasmons. Our metasurface supports dense nanometre-sized gaps as singularities, which are introduced by applying an appropriate mechanical loading to create additional nanometre-sized cracks in the fragment with saturated transferred stress (Fig. 1b), and/or by using biaxial intergranular fractures to drag the deflected fragment domain towards its neighbours at nanometre-sized spacings (Fig. 1c). See Methods and Supplementary Note 1 for details. The packing density (number of singularities per unit area) is controlled by adjusting the biaxial elongations, as also shown in Supplementary Note 1.

Such a strategy offers several advantages. First, the natural nanometre-sized metasurface singularities, usually difficult to be obtained with traditional top-down nanofabrication methods such as electron-beam lithography, allow stronger field localizations and smaller mode volumes (Fig. 1b-f), boosting the coupling strength. Second, considering these high-density plasmonic hotspots plus the randomness of sharp features (hence the polarization insensitivity; see Supplementary Note 2 for details), the average number of interacting excitons (usually characterized as in-plane dipoles) with the plasmonic mode (see the near field of the gap plasmons in Fig. 1f) is also increased.

To characterize our sample, statistical analysis of the scanning electron microscopy (SEM) images shows that the average number ($\bar{N}$) of nanometre-sized gaps within each unit area (1μm$^2$) reaches 10 (Fig. 1d). We do not analyze the packing of gaps much <3 nm because the SEM has difficulty in accurately capturing such small gaps. A representative SEM image of a 20 nm-thick gold metasurface at the highest packing condition (with the peak $\bar{N}$ in Fig. 1d) is shown in Fig. 1e. The fracture morphology also shows the isotropic property, where the average fragment domain size is almost the same in different directions; see detailed geometrical and corresponding optical isotopic properties in Supplementary Note 2. The large interaction between the fragments adjacent to the singularity produces a maximum electric field enhancement of around 100 (Fig. 1f; the colour bar used is to clearly show the electric field at all hotspots) within the small mode volume ($V = 2.03 \times 10^4$ nm$^3$). A pronounced Rabi splitting (blue and purple lines, Fig. 1g) is thus observed, with the extracted normalized coupling strength $\eta = 0.12$ (purple line, Fig. 1g) and an estimated number of 23.6 excitons coherently contributing to the interaction with the surface plasmon, demonstrating USC with the WS$_2$ monolayer.

### Ultrastrong exciton-plasmon coupling and its tunability

The normalized dark-field scattering response of our samples with various gold film thicknesses (ranging from 17 – 24 nm to selectively tune the plasmonic resonance to the WS$_2$ exciton energy, see Supplementary Fig. 10 for details) are plotted in Fig. 2a, b, where the samples are under no strain and −2% uniaxial strain (See Supplementary Fig. 7 for details of the experimental setup to apply uniaxial strain), respectively. Herein, to quantify the coupling strength, we extract the polariton energies (guided by the blue and purple curves in Fig. 2a, b) from the scattering peaks in the spectra and obtain the vacuum Rabi frequency by fitting the spectrum to a coupled oscillator model[32] (See Supplementary Note 4 for details). The extracted characteristic polariton dispersion consists of lower ($\omega_-$) and upper ($\omega_+$) polariton branches (Fig. 2c), where the polariton energies are fitted as the eigenvalues of the full Hopfield Hamiltonian[33,34], yielding

$$(\omega^2 - \omega_{ex}^2)(\omega^2 - \omega_{pl}^2) - 4g^2\omega^2 = 0 \tag{1}$$

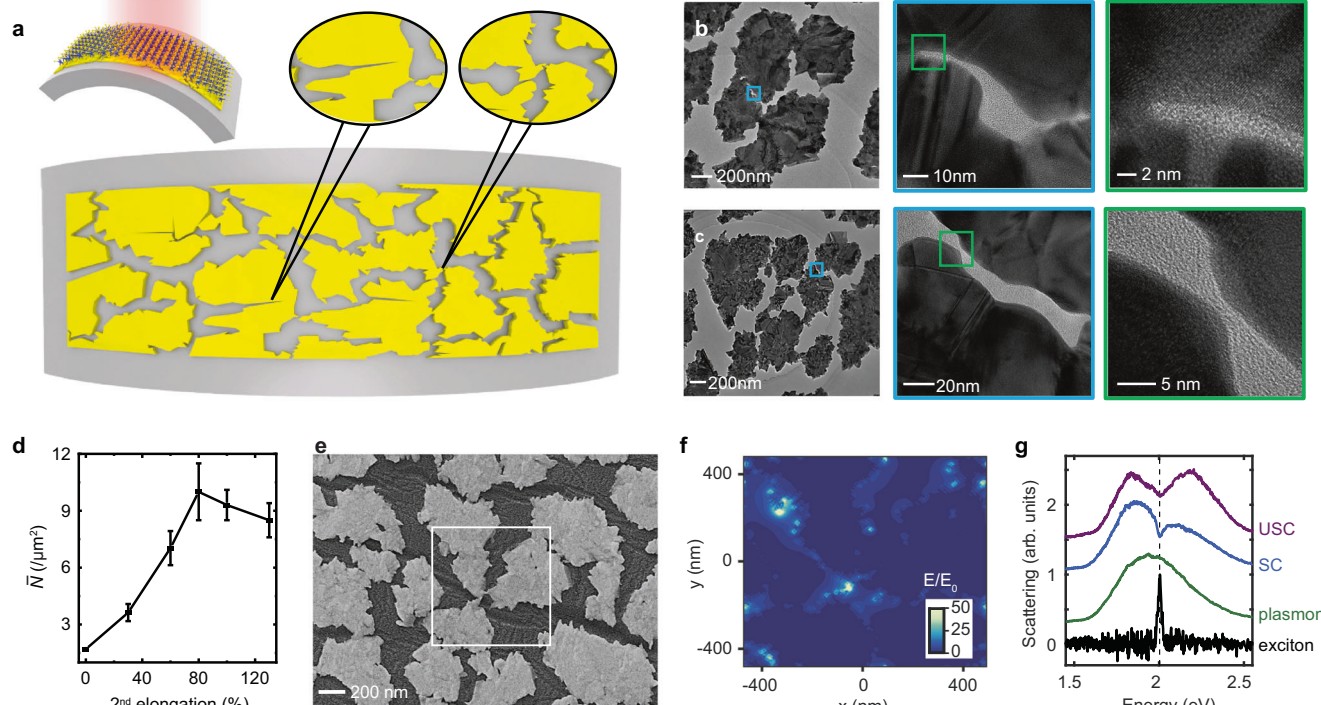

**Fig. 1 | Ultrastrong coupling in the WS$_2$ monolayer with a random multi-singular plasmonic metasurface. a** Schematic of a gold multi-singular metasurface with a dense array of nanometre-sized plasmonic gaps. Insets: left: schematic of a WS$_2$ monolayer integrated with the multi-singular metasurface where the WS$_2$ monolayer was mechanically exfoliated onto a PDMS tape and transferred onto the gold nanopattern using a dry transfer method; middle and right: two main paths for generating nanometre-sized gaps. Middle: the first path, a nanometre-sized crack generates in the fragment with saturated transferred stresses. Right: the second path, adjacent fragment domains are dragged infinitely close together (at nanometre-sized spacings) by biaxial mechanical loadings. **b**, **c** Transmission electron microscopy (TEM) images of a 20 nm-thick gold multi-singular metasurface showing a dense array of nanometre-sized plasmonic gaps. The darker grey areas correspond to gold, and the lighter ones to air. **b** refers to the first path, and (**c**) to the second. **d** Average number ($\bar{N}$) of nanometre-sized (~sub-3 nm in the

scanning electron microscopy image) plasmonic gaps per 1μm² in the 20 nm-thick metasurface as a function of the second (2nd) elongation. Error bars are standard errors from multiple samples. **e** Scanning electron microscopy images of the 20 nm-thick metasurface at 80% 2nd elongation. Note that despite the random distribution and orientation of the singularities, the fabricated multi-singular metasurface retains 'global uniformity', which refers to the uniform distribution of singularities within the laser beam area or over the entire patterned gold film. This global uniformity can be precisely controlled during the cold-etching process (see Supplementary Fig. 6 for details). The white box corresponds to the simulation area in (**f**). **f** The simulated near field of the gap plasmons. **g** Dark-field scattering spectra of the WS$_2$ monolayer on polymer (thermoplastic polyurethane, black) and the plasmonic metasurface uncoupled (green) and coupled (blue for SC and purple for USC) to WS$_2$ excitons in arbitrary units (arb. units).

where counter-rotating and photon self-interaction terms are included. Here, $\omega_{ex}$ and $\omega_{pl}$ are the energies of the WS$_2$ excitons and the plasmonic mode, respectively, and $g$ is the exciton-plasmon coupling strength. $\omega_{pl}$ is calculated from the extracted $\omega_{\pm}$ as $\omega_{pl} = \omega_{+}\omega_{-}/\omega_{ex}$. The fitting leads to the typical anticrossing behaviour, with $g = 165.9$ meV for the no-strain case and $g = 240.4$ meV for the −2% uniaxial strain, corresponding to a normalized coupling strength of η = 0.082 and enhanced η = 0.12, respectively (Fig. 2c). This clearly indicates that the system operates in the USC regime when uniaxial strain is applied. Similar normalized coupling strengths are observed in different batches of samples (see Supplementary Figs. 9 and 12), demonstrating the robustness of our platform for observing USC. The normalized coupling strength in the WS$_2$ monolayer coupled to the multi-singular metasurface dramatically exceeds that of the reported optimized conventional TMD monolayer in plasmonic systems (Supplementary Fig. 13, measured under exactly the same experimental conditions), suggesting that the coherent interaction of multiple excitons with the nanometre-sized plasmonic hotspots supported by the metasurface is the key factor in achieving the observed high coupling strength. Note that the observed Rabi splitting is an average value over an ensemble of detuned WS$_2$ exciton-plasmon coupling subsystems within the laser beam. Furthermore, the SC condition (i.e., coupling strength exceeds loss) $g > (\gamma_{pl} + \gamma_{ex})/4$ is satisfied, where the damping

losses of plasmonic resonance and exciton emission are $\gamma_{pl} = 380$ meV and $\gamma_{ex} = 45$ meV, respectively.

Applications of USC to nonlinear optics include low-threshold frequency conversion[35] and phase-matched optical amplification[36]. To demonstrate the advantages of our system with tunable USC, here we exploit USC towards tunable resonant polariton-enhanced nonlinearity. It is known that the excitonic resonances in monolayer TMDs can facilitate second harmonic generation (SHG)[37]. In our system, USC allows spectral splitting of the polaritonic resonance with tunable ground state energies and, consequently, leads to dispersive polariton-enhanced SHG in the WS$_2$ monolayer. Strong evidence for the pronounced SHG intensity enhancement and spectral splitting can be seen from the nonlinearity spectrum as a function of the pump wavelength under two different values of the applied strain (Fig. 2d). As can be seen in Supplementary Fig. 14, the polariton-enhanced SHG from the WS$_2$ monolayer is around 15 times stronger than that on PDMS, and polarization-independent polariton-enhanced SHG is observed due to the isotropy of our multi-singular metasurface.

Furthermore, thanks to the flexible nature of the TPU polymer substrate, the mechanical bending in our system can be adjusted to modify the plasmonic gap size and the corresponding packing density for modulating exciton-plasmon coupling strength. Under upward (tensile strain) or downward (compressive strain) bending, both the

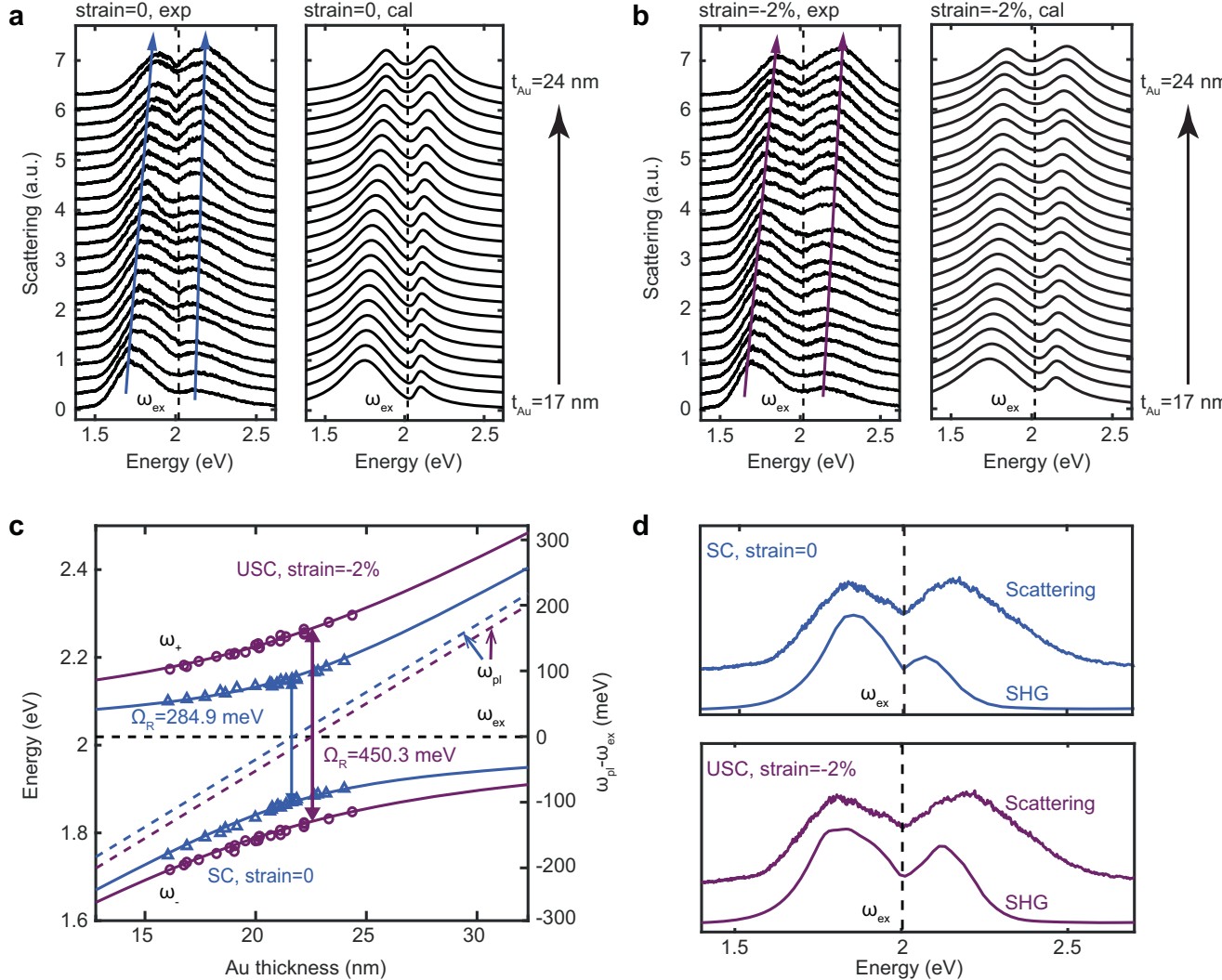

**Fig. 2 | Ultrastrong exciton-plasmon coupling in the WS₂ monolayer.** Dark-field scattering spectra under (**a**) 0 and (**b**) −2% uniaxial strain on the metasurface over different gold film thicknesses (from bottom to top, the gold film thickness ($t_{Au}$) increases from 17 nm to 24 nm). The scattering splits into lower and upper polariton branches, exhibiting level anticrossing. The vertical dashed lines refer to the WS₂ exciton energy ($\omega_{ex}$ = 2.019 eV). The right panels are calculated scattering spectra using the model of two coupled oscillators. **c** Dispersion plots of the measured dark-field scattering spectra. The lower ($\omega_-$) and upper ($\omega_+$) polariton branches are extracted from the scattering spectra in **a,b** and fitted (solid lines) with a coupling strength of 165.9 meV at 0 strain (blue lines) and 240.4 meV at −2% strain (purple lines) in the full Hopfield Hamiltonian. The experimental spectral peaks are shown as triangles (0 strain) and circles (−2% strain). The exciton energy ($\omega_{ex}$) is shown as the horizontal dashed black line. The plasmonic mode energy

($\omega_{pl}$) is shown as the diagonal dashed blue (SC) and purple (USC) lines. The plasmonic response shifts to lower energy when compressive strain is applied, and the local field enhancement is largely enhanced due to the reduced gap size, and both effects lead to significant changes in the polaritonic dispersion. **d** Normalized second harmonic generation (SHG) intensity compared to scattering in the SC (upper, strain = 0) and USC (lower, strain = −2%) regime, respectively. The simultaneous emergence of energy splitting in dark-field scattering, photoluminescence (Supplementary Fig. 15) and SHG spectra precludes Fano interference phenomena from being responsible for the observed anticrossing. Note that the deviation between SHG and dark-field scattering splittings most likely arises from measurement errors in the strength of the SHG signal at the spectral edge of the photon detector.

plasmonic resonance (left, Fig. 3a) and the corresponding plasmonic enhancement in photoluminescence peak intensity (right, Fig. 3a) can be tuned. We notice that the small strain applied in this work has a negligible effect on the exciton energy (Supplementary Fig. 16), because the WS₂ monolayer is loosely contacted with the surface of the plasmonic metasurface. Dark-field scattering spectra at different strains are rendered in Fig. 3b, with two polariton branches well fitted (black solid lines, Fig. 3b) by the full Hopfield Hamiltonian described above. Specifically, the normalized coupling strength varies gradually from 0.075 – 0.12 (over the strain range from +0.375% − −2%, Fig. 3c), clearly demonstrating the tunability of USC. We expect even higher coupling strengths by further increasing the compressive strain, but to preserve the resilience and robustness of the flexible substrate, we

have avoided attempting to increase the compressive strain beyond −2%.

One interesting implication of the USC is that the global vacuum energy of the system is modified with respect to the coupling strength (Fig. 3d) by dressing excitons with light[33,38,39]. In our system, the corresponding change in the ground state is calculated as $\Delta E_G = \hbar\left(\omega_+ + \omega_- - \omega_{ex} - \omega_{pl}\right)/2$. Note that the ground state energy of a harmonic oscillator is half that of the transition energy[40]. The normalized ground-state energy variation ($\Delta E_G/E_G$) versus strain was calculated and fitted to the extracted coupling strengths (Fig. 3c) and polariton energies $\omega_\pm$ (Fig. 3b), yielding a modification of 0.67% (Fig. 3d). The absolute ground-state energy modification reaches 13.4 meV at −2% compressive strain.

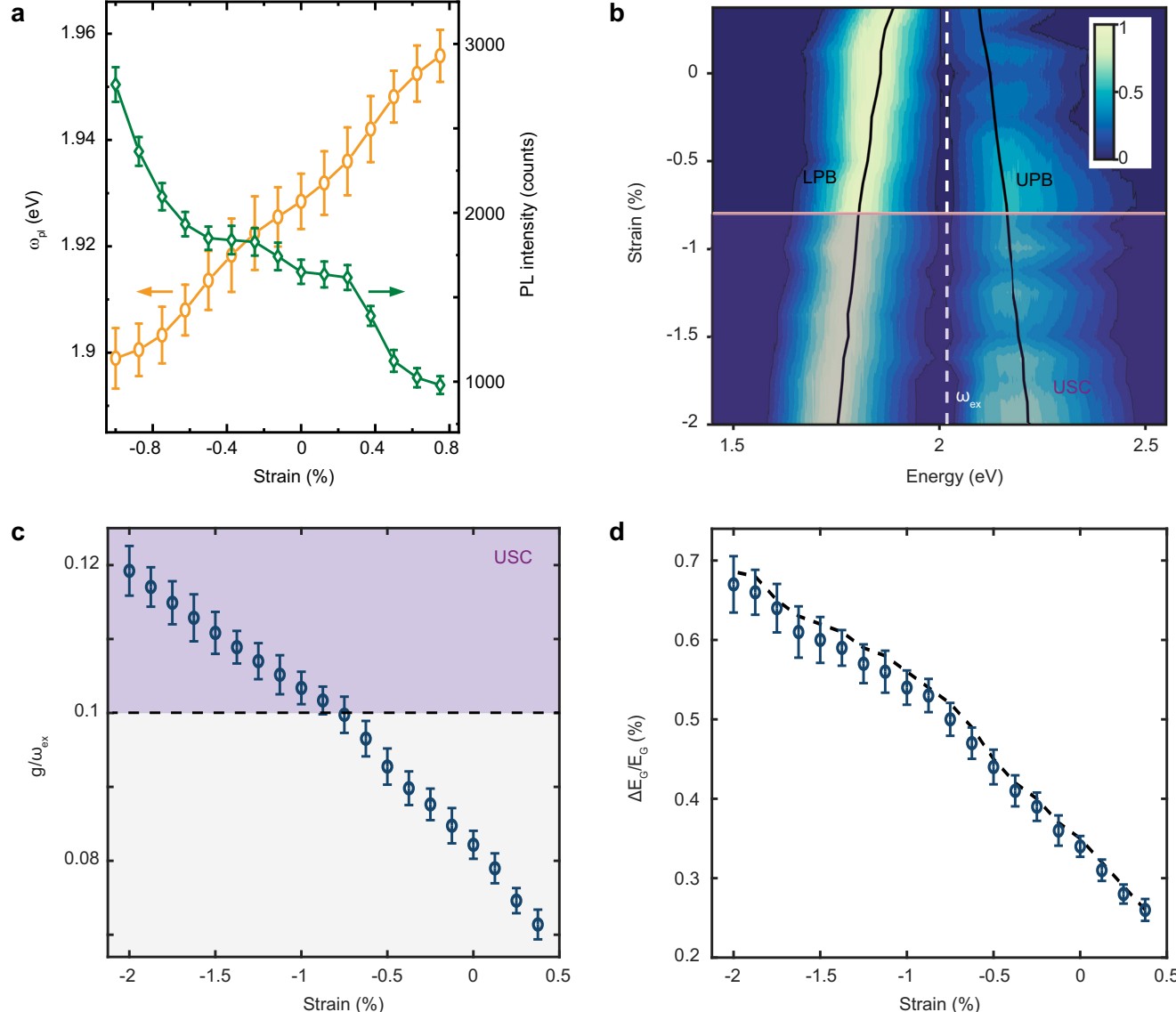

**Fig. 3 | Tunable ultrastrong exciton-plasmon coupling in the WS$_2$ monolayer.** **a** Mechanically tunable plasmonic resonance (energy at $\omega_{pl}$) and plasmonic enhancement in photoluminescence peak intensity of the strongly coupled plasmonic system. **b** Normalized dark-field scattering intensity as a function of excitation energy and strain. The horizontal purple line marks the onset of USC. The black solid lines are the extracted polariton energies of the lower (LPB) and upper (UPB) polariton branches. The white dashed line is the WS$_2$ exciton energy ($\omega_{ex} = 2.019$ eV). The plasmonic response shifts to lower/higher energy when compressive/tensile strain is applied, and the local field enhancement is

significantly enhanced/reduced owing to the reduced/enlarged gap size, and both effects change the **c**oupling strength. **c** Normalized coupling strength as a function of strain. The onset of USC is marked by the horizontal dashed line. **d** Ground-state energy modification as a function of strain. The black dashed line is the calculation result from the coupling strength ($g$) and the polariton energies ($\omega_+, \omega_-$) from the fitting to the full Hopfield Hamiltonian. The absolute change in ground-state energy reaches 13.4 meV at −2% compressive strain. Error bars in (**c**) and (**d**) result from the variation in dark-field scattering from multiple batches of samples and are extracted from the standard error of the fit.

## Ultrastrong coupling in WS$_2$ multilayers

Last, we show that the coupling strength in USC is also relevant to the layer numbers of WS$_2$. We transferred trilayer and quadrilayer WS$_2$ flakes onto several metasurfaces with different gold layer thicknesses (19 nm to 23 nm) on flexible PDMS substrates. The dispersion curves of the scattering spectra are shown in Fig. 4, showing the increased energy splitting between polaritons compared to the monolayer. In trilayer WS$_2$ (Fig. 4a,b), the Rabi splitting ($\Omega_R$) exceeds 397.9 meV ($\eta \sim 0.108$) and 563.9 meV ($\eta \sim 0.147$) under 0 and −2% compressive strain, respectively, while in quadrilayer WS$_2$ (Fig. 4c, d), $\Omega_R$ can be as high as 449.7 meV ($\eta \sim 0.12$) and 634.7 meV ($\eta \sim 0.164$) accordingly. Moving from the monolayer to the quadrilayer cases, the normalized coupling strength increases with the number of layers, but less rapidly than predicted by a square root

function, as shown in Fig. 4e, because the coupling strength depends on both the number of excitons and the spatial overlap of the confined electric field with the WS$_2$ layers (which decreases as the number of WS$_2$ layers increases). Thus, we have successfully combined monolayer and multilayer WS$_2$ flakes with multi-singular metasurfaces to create a large set of ultrastrongly coupled exciton-plasmon systems under ambient conditions. Moreover, the cooperativity, defined as $C = 4g^2/\gamma_{pl}\gamma_{ex}$ (a key figure of merit to characterize the coupling regimes of light-matter interaction) in our samples outperforms all room-temperature TMD-based platforms ($C = 28.4$ in quadralayer WS$_2$).

We compare our system with the previous reports of light-matter coupling, as summarized in Supplementary Table 1 for details. The normalized coupling coefficients of other strongly coupled systems

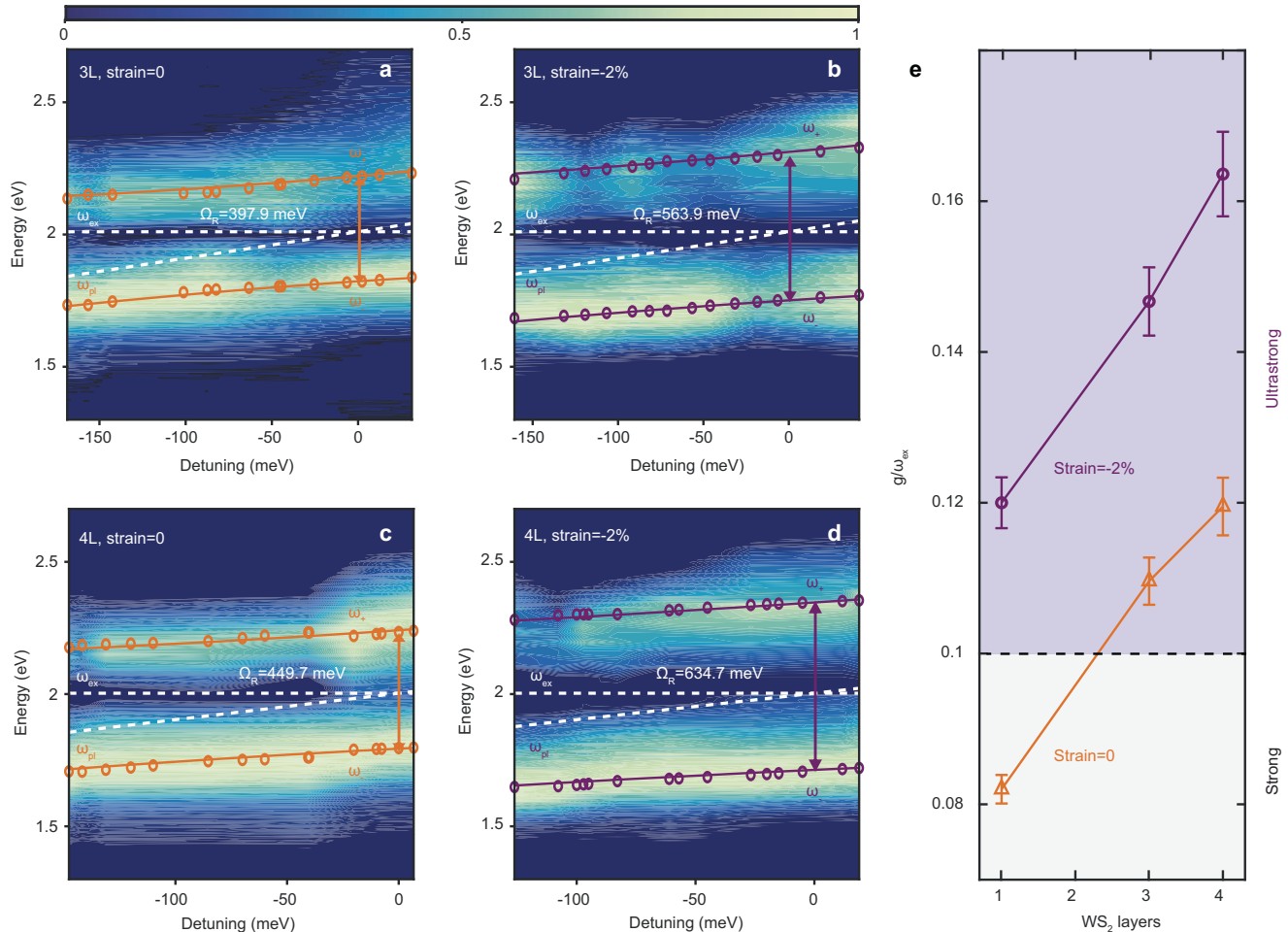

**Fig. 4 | Ultrastrong coupling in multilayer WS₂ flakes. a–d,** Dark-field scattering spectra of a multilayer WS₂ plasmonic system. Dashed white lines denote the exciton energy ($\omega_{ex} = 2.01$ eV) for **a,b** $\omega_{ex} = 2.003$ eV for **c,d** and the diagonal dashed orange and purple lines indicate the plasmon energy ($\omega_{pl}$). The open circles represent lower ($\omega_-$) and upper ($\omega_+$) polariton energies obtained from the dark-field scattering spectra of individual plasmonic systems. The solid lines show the lower and upper polariton dispersions using the full Hopfield Hamiltonian. 3 L and 4 L represent the trilayer and quadrilayer cases. **e** Normalized coupling strength ($g/\omega_{ex}$) as a function of the WS₂ layers under 0 (orange) and −2% compressive strain (purple). The horizontal dashed line marks the onset of USC. Error bars are derived from the variation in dark-field scattering from different batches of samples and are extracted from the standard error of the fit.

are typically in the range of 0.015–0.03 (0.015–0.04) for monolayer[29–31] (multilayer[27,28,41,42]) TMDs, 0.02–0.095 for organic molecules (<20 nm-thick)[43–45], and 0.04–0.09 for quantum dots[46–48]. Existing room-temperature USC systems are typically implemented using organic molecules, and the reported thickness must be >60 nm[18,49,50]. Landau polaritons and superconducting circuits are not included in this comparison because their transition energies are far below those of TMD excitons. Our platform for USC, with an emitter thickness of <1 nm, is thus unique and holds great promise for next-generation atomically thin optoelectronics in the visible.

## Discussion

In summary, we have shown the ultrastrong exciton-plasmon coupling at room temperature in WS₂ multilayers coupled to random multi-singular plasmonic metasurfaces formed by cold-etching, which can be further tuned by mechanical strain in a flexible substrate. Our results promise the USC in the atomic layer limit under ambient conditions, which could be further extended to scenarios of different metals or doped semiconductors and complex 2D heterostructures for more exotic complex light-matter interactions, such as tunable trion polaritons, van der Waals heterostructure polaritons, and moiré-induced optical nonlinearities. Our work could also lead to new insights in fundamental science and potential applications in

the fields of nonlinear nanophotonics and analytical chemistry, among others.

## Methods
### Experimental structure

WS₂-coupled multi-singular plasmonic metasurface: A thin gold film was deposited on an amorphous poly(etherimide) (PEI) polymer substrate (125 μm-thick) using an electron beam evaporator at a deposition rate of 2 Å s⁻¹. During the evaporation, the temperature of the vacuum chamber was maintained below 60 °C throughout the process to prevent thermal expansion or deformation of the PEI film due to the build-in stress, which would cause wrinkles or defects in the gold film. For the cold-etching process, the first stretch was performed by stretching the gold/PEI film in the x-direction and stopping when the necks extended along the length of the film. The second stretch was performed by re-stretching the as-fabricated film in the y-direction and the elongation extent was controlled to obtain gold nanopatterns with different second elongations. After the cold-etching process, the resulting gold nanopatterns were transferred to a flexible TPU substrate (2 mm thick) by a dry peel-off method. WS₂ monolayers and multilayers were mechanically exfoliated from the commercial bulk crystals onto a PDMS tape and transferred to the gold multi-singular metasurface (on TPU) through a dry transfer method (see Supplementary Fig. 1 for details).

## Optical characterizations

All optical characterizations were performed in reflective geometry at room temperature. For all linear polarization measurements, the metasurface system was rotated, and the polarization-dependent components were analyzed. For photoluminescence measurements, a 532 nm diode laser was used to excite the sample. A 100 × microscope objective lens (numerical aperture (NA) D 0.75) was used and the incident laser power was ~100 μW with a laser spot size of ~4 μm. Dark-field scattering (see Supplementary Fig. 8 for the setup) measurements were carried out using a hyperspectral imaging system with a broadband halogen lamp as the light source. A 50 × objective (NA D 0.55) was used, and the incident light power was ~20 μW with a laser spot size of ~5 μm.

## Numerical calculations

Commercial finite-difference time-domain software was used to calculate the field enhancement of the gap plasmons. The permittivity $WS_2$ is modelled as a Lorentzian oscillator $\varepsilon(\omega) = 1 + \sum_{k=1}^{N} f_k / (\omega_k^2 - \omega^2 - i\gamma_k\omega)$, with $f_k$, $\gamma_k$, and $\omega_k$ being the oscillator strength, the linewidth of the $k$th oscillator, and the oscillation energy, respectively. The permittivity of the gold was from the software database.

## Hopfield Hamiltonian

The Hopfield Hamiltonian of our plasmonic system in the USC regime contains three main blocks:

$$H = H_{sys} + H_{int} + H_{A^2} \tag{2}$$

with the Hamiltonian of the closed-plasmonic system

$$H_{sys} = \hbar\omega_{ex}\hat{a}^\dagger\hat{a} + \hbar\omega_{pl}\hat{b}^\dagger\hat{b} \tag{3}$$

and the interaction Hamiltonian

$$H_{int} = i\hbar g(\hat{a}^\dagger + \hat{a})(\hat{b} - \hat{b}^\dagger) \tag{4}$$

and the photon self-interaction ($A^2$) Hamiltonian

$$H_{A^2} = \frac{\hbar g^2}{\omega_{ex}}(\hat{a}^\dagger + \hat{a})(\hat{a}^\dagger + \hat{a}) \tag{5}$$

$\hat{a}^\dagger$ and $\hat{a}$ are the exciton creation and annihilation operators, respectively, and $\hat{b}^\dagger$ and $\hat{b}$ are those of the localized plasmons. Since $H$ is invariant under translation, we define

$$\hat{c} = w\hat{a} + x\hat{b} + y\hat{a}^\dagger + z\hat{b}^\dagger \tag{6}$$

And we can get

$$[\hat{c}, H] = E\hat{c} \tag{7}$$

The eigenvalue problem is rewritten in matrix form,

$$\begin{bmatrix} \omega_{pl} + 2g^2/\omega_{ex} - i\gamma_{pl}/2 & -ig & -2g^2/\omega_{ex} & -ig \\ ig & \omega_{ex} - i\gamma_{ex}/2 & -ig & 0 \\ 2g^2/\omega_{ex} & -ig & -\omega_{pl} - 2g^2/\omega_{ex} + i\gamma_{pl}^*/2 & -ig \\ -ig & 0 & ig & -\omega_{ex} - i\gamma_{ex}^*/2 \end{bmatrix} \begin{bmatrix} w \\ x \\ y \\ z \end{bmatrix} = E \begin{bmatrix} w \\ x \\ y \\ z \end{bmatrix} \tag{8}$$

The two eigenvalues $\omega_{\mp}$ of the above matrix are the positive solutions of the equation[30,45,46],

$$(\omega^2 - \omega_{ex}^2)(\omega^2 - \omega_{pl}^2) - 4g^2\omega^2 = 0 \tag{9}$$

from which we obtain

$$\omega_+\omega_- = \omega_{pl}\omega_{ex} \tag{10}$$

To interpret our experimental data in the SC regime, we obtain the analytical polariton dispersion energies with a coupled oscillator model with the Hamiltonian

$$H = \begin{bmatrix} \omega_{pl} - i\gamma_{pl}/2 & g \\ g & \omega_{ex} - i\gamma_{ex}/2 \end{bmatrix} \tag{11}$$

as

$$\omega_\pm = \frac{1}{2}\left(\omega_{pl} - i\frac{\gamma_{pl}}{2} + \omega_{ex} - i\frac{\gamma_{ex}}{2}\right) \pm \sqrt{g^2 + \frac{1}{4}\left(\omega_{pl} - i\frac{\gamma_{pl}}{2} - \omega_{ex} + i\frac{\gamma_{ex}}{2}\right)^2} \tag{12}$$

Note that the complex Rabi splitting can be calculated as $\Omega_R = \omega_+ - \omega_- = 2\sqrt{g^2 + \frac{1}{4}\left(\omega_{pl} - i\frac{\gamma_{pl}}{2} - \omega_{ex} + i\frac{\gamma_{ex}}{2}\right)^2}$, where for zero detuning and zero damping the rabi splitting is $\Omega_R = 2g$.

## Strain model

The Young's modulus, ultimate strength and thickness of the gold film are given by $E$, $\sigma_s$ and $h$, respectively. Assume that the interfacial shear stress is a constant $\tau_0$ for any interfacial displacement and the substrate stretching is accompanied by a steady neck propagation with the necked stretch ration of $\lambda_n$.

The gold film fractures sequentially into fragments, following the propagating neck. For a fragment of film, its size in the initial configuration (undeformed configuration) can be determined as

$$L_0 = \frac{Eh}{\tau_0}\left(1 - e^{-\sigma_s/E}\right). \tag{13}$$

The portion of substrate with the same initial position and length as the film fragment experiences full necking and now has the size

$$L_{subs} = \lambda_n L_0 = \frac{\lambda_n Eh}{\tau_0}\left(1 - e^{-\sigma_s/E}\right). \tag{14}$$

Assume that the residual stress in the film fragment is distributed linearly within the fragment and has the maximum value of $\sigma_r$. Then we have

$$L_0 = \frac{2Eh}{\tau_0}\left(1 - e^{-\sigma_r/E}\right). \tag{15}$$

With Eqs. (13) and (15), we obtain

$$\sigma_r = E \ln \frac{2}{1 + e^{-\sigma_s/E}}. \tag{16}$$

Therefore, the size of the film fragment after rupture is

$$L_{film} = \frac{2\sigma_r h}{\tau_0} = \frac{2Eh}{\tau_0} \ln \frac{2}{1 + e^{-\sigma_s/E}}, \tag{17}$$

and the distance between neighbouring film fragments, i.e., the averaged gap size in the metasurface, can be expressed as

$$\bar{L}_{gap} = L_{subs} - L_{film} = \frac{Eh}{\tau_0} \left( \lambda_n - \lambda_n e^{-\sigma_s/E} - 2\ln\frac{2}{1 + e^{-\sigma_s/E}} \right). \tag{18}$$

If we assume that the experimental gap size follows a normal distribution, $L_{gap} \sim N(\mu, \sigma^2)$, where $\mu = \bar{L}_{gap}$ and the variance $\sigma^2$ can be obtained from experimental measurements. The number of nanometre-sized gaps ($L_{nano}$) can be obtained from the normal distribution by determining the area satisfying $L_{gap} - \mu < L_{nano}$.

## Data availability

The data supporting the current study in the paper are included in the paper and/or the Supplementary Materials. Additional data related to this paper can be requested from the corresponding authors.

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

## Acknowledgements

This work was supported by the Singapore National Research Foundation Competitive Research Program (NRF-CRP22-2019-0006, NRF-CRP23-2019-0007, NRF-CRP22-2019-0064 and NRF-CRP22-2019-0007), the Singapore Ministry of Education Academic Research Fund Tier 2 (MOE2019-T2-2-127, MOE-T2EP50120-0002, MOE-T2EP50120-0009, MOE-T2EP50220-0020 and MOE-T2EP50122-0005), AcRF Tier 1 (RG57/21, RG156/19 (S)), AcRF Tier 3 (MOE2016-T3-1-006 (S)), A*STAR (1720700038, A1883c0002, A18A7b0058, A20E5c0095, A2083c0062 and A2090b0144), Spanish Ministry for Science and Innovation-Agencia Estatal de Investigación (AEI) through grants CEX2018-000805-M and PID2021-125894NB-I00, the Autonomous Community of Madrid, the Spanish government and the European Union through grant MRR Advanced Materials (MAD2D-CM) National Medical Research Council (NMRC) (021528-00001), A*STAR IAF-ICP Programme I2001E0067, the Schaeffler Hub for Advanced Research at NTU, and the Distinguished Professor Fund of Jiangsu Province (Grant No. 1004-YQR24010).

## Author contributions

T. Wu and Y. Luo designed the research project. T. Wu and M. Chen fabricated the flexible plasmonic metasurfaces; C. Wang fabricated the $WS_2$ flakes and the bowtie/dimer antennas. T. Wu, L. Liu, J. Zhao, and K. Chaykun conducted the optical experiments. D. Li developed the strain model. Z. Wang, Z. X. Wang, and T. Wu took the SEM and TEM images. T. Wu, G. Hu and L. Liu performed the finite-difference time-domain simulations and theoretical analysis. S. Zhu and T. Wu conducted the lifetime measurement. F. J. Garcia-Vidal, L. Wei, Q. Wang, and Y. Luo supervised the research. T. Wu, G. Hu, and L. Liu analyzed the data; T. Wu and G. Hu wrote the manuscript. Q. Xiong, Z. Shen, H. Gao, and all other authors contributed to data interpretation and editing the manuscript.

## Competing interests

The authors declare no competing interests.
