## [Peer Review File · Nature Communications]

Ultrastrong exciton-plasmon couplings in WS₂ multilayers synthesized with a random multi-singular metasurface at room temperatureREVIEWER COMMENTS

Reviewer #1 (Remarks to the Author):

This manuscript reports a ultrastrong exciton-plasmonic coupling in WS₂ layers at room temperature, meanwhile the phenomenon can be tuned by strain due to the structure prepared on a flexible polymer substrate. It will be worthy of consideration for publication if the following questions are solved in the manuscript.

Specific comments are:

1. Although the method avoids to seek perfect metals or high-quality resonators, the sample with ultrastrong coupling is completely random due to uncontrollable gold film thickness and metasurface. The well-designed Au antenna or array structure should be made to obtain a stable ultrastrong coupling. So, I suggest that authors try to find the physical rules and optimize random multi-singular metasurface.
2. The lifetime of exciton-plasmon polariton need to be measured to demonstrate the difference of the ultrastrong coupling and strong coupling. Furthermore, the lifetime of exciton-plasmon polariton need to be measured under different strain.
3. How to fit the data in Fig 2.a and b to obtain the coupling strength in Fig 2c? A detailed description is necessary, and the fitting process requires Au thickness but this parameter does not appear in equations 10 and 12.
4. Despite the explanation given in Fig. 3a, the mechanism for strain tuning the ultrastrong is not described. The compressive and tensile strain deviate the plasmonic resonance wavelength from the exciton wavelength in WS₂, but the compressive strain leads to the coupling strength enhancement, which is opposite of the tensile strain. Please explain in detail the mechanism of coupling strength change under different compressive and tensile strain in Fig 3c.
5. In Fig. S1, the schematics should be numbered and organized so that the reader can understand them. In additional, TPU, PDMS and PEI should be clearly annotated, Otherwise, it's easy to confuse the reader.
6. The ultrastrong coupling maybe led to occur Bose-Einstein condensation (BEC) phenomenon at low temperature, Is it possible that doing experiments at low temperature?
7. The equipment and method of the strain application should be described in the supplementary material, to help readers to understand the process. Furthermore, the schematic of dark field scattering spectra under strain should be shown in the supplementary material to the reader understand this test method.

Reviewer #2 (Remarks to the Author):

Light-matter interaction has been an important topic for more than a hundred year. Recently, in the quantum regime, ultrastrong coupling is highly demanded. In this work, the authors fabricated a metasurface with many nanometre-sized plasmonic hotspots to which WS₂ excitons coherently interact, achieving the ultrastrong exciton-plasmon coupling. This work is interesting. Before making the final recommendation, I have a couple of questions.

1. Usually, ultrastrong coupling can overcome the disturbance caused by the linewidth. Here, the linewidth is still very broad. Please comment on this issue and its influence on the significance of ultrastrong coupling, especially its possible application in quantum optics.
2. From Fig. 2a-b to Fig. 2c, how did the authors extract the resonant frequencies of the two branches? Please show the details, since the two original resonances have significantly different linewidth.
3. For each measurement, the light spot would cover a wide range of hotspots, i.e., the LSPR peak here is from an ensemble of nanogaps. Did the author measure the fluctuation of the LSPR peak of each sample? If so, at least one example should be shown, which is the scattering spectra from different positions of a single sample. Also, a standard error should be given. Only in this way, the potential readers could replicate all the experiments and data.
4. Please comment on the fact in Fig. 2d: the splitting of SHG spectra is not as large as that of the scattering spectra.
5. Fig. S15 is not cited in the main text. Please further comment on the mechanism why the upper branch is not shown in PL spectra.
6. Grammar and format.

Reviewer #3 (Remarks to the Author):

Strong coupling between 2D exciton and surface plasmon has been intensively studied in the past few years and has resulted in a number of published papers. In this work, the authors report on the observation of ultrastrong coupling of surface plasmon and exciton in two-dimensional van der Waals semiconductor (monolayer and multilayer tungsten disulfide). This coupling is realized by formation of random plasmonic “hot gaps” via etching of a thin gold film deposited on a flexible polymer substrate. The main differences from published works are the observation of large Rabi splitting (300 – 450 meV) and mechanical tunability due to the use of a flexible substrate. However, there are two obvious drawbacks: (1) Their approach is stochastic in nature and difficult to have real applications (e.g., lasing, nonlinear optics claimed by the authors), (2) ultrastrong coupling (large Rabi splitting) in a plasmonic platform has previously been reported. My evaluation is to enhance the technical contents and publish

the revised paper in a regular journal, since this work does have the novelty and urgency to warrant publication in Nature Communications.

Two suggestions for revision:

1. The imaginary part (linewidth) evolution in the ultracoupling region should be analyzed and presented, especially for a coupling system with large difference in linewidth (380 meV vs. 45 meV).
2. It is necessary to provide the detailed information about coupling of exciton and plasmon in the case of multilayer WS₂ flake, since intra and interlayer excitons have different dipole orientations and different coupling strengths with plasmon in “hot spots”.

Response to Reviewers' Comments:

We would like to thank all the reviewers for their careful reviewing of our work and for their valuable suggestions. We are also grateful for their inspiring comments, which have allowed us to further improve the manuscript. Below, we provide a detailed point-by-point response which we believe addresses each reviewer's questions and comments. The reviewers' comments are in *italic*, and the authors' responses are in **Roman**. All changes in the revised manuscript are marked in **red**.

Referee #1:

This manuscript reports a ultrastrong exciton-plasmonic coupling in WS₂ layers at room temperature, meanwhile the phenomenon can be tuned by strain due to the structure prepared on a flexible polymer substrate. It will be worthy of consideration for publication if the following questions are solved in the manuscript.

Response:

We really appreciate your thorough review of our work. We are thankful for your inspiring comments, which have allowed us to further improve the manuscript, in terms of both the theoretical and the experimental aspect. In the following, we provide a detailed point-by-point response to your questions and comments and describe the corresponding changes in both the main text and the Supplementary Information.

Comment #1:

Although the method avoids to seek perfect metals or high-quality resonators, the sample with ultrastrong coupling is completely random due to uncontrollable gold film thickness and metasurface. The well-designed Au antenna or array structure should be made to obtain a stable ultrastrong coupling. So, I suggest that authors try to find the physical rules and optimize random multi-singular metasurface.

Response:

We thank the referee for this valuable suggestion. We would like to clarify that the gold thickness is well controlled during the e-beam deposition process, where the thin gold films are deposited on the poly(ethyleneimine) (PEI) substrate at a rate of 0.2 Ås⁻¹ by e-beam evaporation.

The complete strain model, as detailed in the second part of the Methods section of the original manuscript, serves as a comprehensive guide for the preparation of our random multi-singular plasmonic metasurface exhibiting a controllable fracture morphology of the metal film, and thus allows us to achieve controlled plasmonic gaps in terms of the gap size. With this strain model, by selecting an appropriate thermoplastic polymer (such as PEI in our case) for implementing the cold-drawing process to fracture the thin metal film^{1,2}, we can determine both the average gap size and the density of the nanometer-sized plasmonic gaps based on the metal categories and their thicknesses.

To illustrate the well-controlled nature of our multi-singular plasmonic metasurface, we have added 'global uniformity' to the caption of Fig. 1e in the revised manuscript to represent the macroscale reliability, stability and repeatability of the random multi-singular metasurface. The 'global uniformity' refers to the uniform distribution of singularities within the laser beam area or across the entire patterned gold film, while 'random' refers to the random distribution and orientation of singularities within the nanoscale area. We highlight that the global uniformity plays a key role to enhance, for example, the light-matter interaction, SERS, nonlinear effects, etc. Hence, to demonstrate the capability of our multi-singular metasurfaces in controlling global uniformity, we have provided reproducible measurements of the dark-field scattering spectra of a 20 nm thick metasurface with and without a monolayer WS₂ on top (Fig. R1a,b, which is Fig. S10a,b in the revised Supplementary Information). Parallel characterizations are performed at 10 different random positions over the corresponding plasmonic system. The relative standard deviations of the resonance wavelength, bandwidth, and the Rabi splitting were found to be 6.2%, 8.2% and 7.5%, respectively, confirming the stability and repeatability of our random multi-singular metasurface. Furthermore, no significant difference in the scattering spectra was observed when the measurements were repeated 12 months after the samples were fabricated (Fig. R1c, which is Fig. S10c in the revised Supplementary Information), further demonstrating the stability and robustness of the plasmonic metasurfaces.

Thus, our proposed fabrication method, despite yielding a random distribution of singularities, provides a reliable and stable fabrication process for producing controllable and repeatable random multi-singular metasurfaces. We do agree with the reviewer that there is still space for us to improve the proposed fabrication method to obtain better organized nanoantennae and array structures. We will leave such an investigation for our future work.

Changes introduced in the revised version:

- 1) In the revised main text, we have added 'Note that despite the random distribution and orientation of the singularities, the fabricated multi-singular metasurface retains 'global uniformity', which refers to the uniform distribution of singularities within the laser beam area or over the entire patterned gold film. This global uniformity can be precisely controlled during the cold-etching process (see Fig. S7 in SI for details).' to the caption of Fig. 1e.
- 2) In the revised main text, we have added 'If we assume that the experimental gap size follows a normal distribution, $L_{\text{gap}} \sim N(\mu, \sigma^2)$, where $\mu = \bar{L}_{\text{gap}}$ and the variance σ^2 can be obtained from experimental measurements. The number of nanometre-sized gaps (L_{nano}) can be obtained from the normal distribution by determining the area satisfying $L_{\text{gap}} - \mu < L_{\text{nano}}$.' to the end of the '**Strain model**'

in the Methods section. Such a description reflects that the strain model provides a concrete physical guide to determine not only the average size of the gap, but also the density of the nanometre-sized gap.

- 3) In the revised Supplementary Information, we have added Fig. R1 as Fig. S10 to show the global uniformity in detail.

Fig. R1| Dark-field scattering spectra measured at 10 different random positions of a 20 nm thick Au metasurface a, without and b, with monolayer WS₂ on top. The relative standard deviation values at the resonance wavelength/bandwidth of the bare plasmons and the Rabi splitting are 6.2%/8.2% and 7.5%, respectively, indicating the stability and repeatability of our random multi-singular metasurface. c, Dark-field scattering spectra at 6 different random positions of a 20 nm thick Au metasurface. Black lines correspond to the signal of a newly prepared sample, blue lines correspond to the signal repeated 12 months after the samples were fabricated at the same sample positions. No significant difference in the scattering spectra when the measurements were repeated 12 months after the samples were fabricated, further demonstrating the stability and robustness of the plasmonic metasurfaces.

Comment #2:

The lifetime of exciton-plasmon polariton need to be measured to demonstrate the difference of the ultrastrong coupling and strong coupling. Furthermore, the lifetime of exciton-plasmon polariton need to be measured under different strain.

Response:

We thank the referee for this constructive comment. The ultrafast dynamics of both uncoupled and coupled systems were investigated at room temperature using pump-probe transient absorption spectroscopy with pumping at 590 nm. In Fig. R2, the differential transition ($\Delta T/T$) spectra from the WS₂ monolayer, bare Au metasurface, and strong coupled and ultrastrong (at -2% compressive strain) coupled plasmonic systems are presented across two time regimes: (a) -0.5 to 7 ps; (b) -0.2 to 2 ps. The intrinsic radiative lifetime of WS₂ is observed to be 250 ± 50 fs, while the lifetime of the Au metasurface is determined to be 1.8 ± 0.05 ps. For the strong (ultrastrong) coupled plasmonic system, the

lifetime value is 60 ± 5 (55 ± 3) fs, indicating a radiative decay rate approximately 4.2 (4.6) times faster than that in the WS_2 monolayer and 30 (32.8) times faster than that in the Au metasurface.

The similarity in lifetimes for SC and USC is attributed to the relatively large beam sizes of the pump and probe pulses used in our measurements, approximately $100 \mu\text{m}$ and $80 \mu\text{m}$ respectively. In a pump-probe system, a large laser beam size is essential to achieve high pump pulse intensity, which improves the signal-to-noise ratio, facilitates alignment, reduces the sensitivity to changes or fluctuations in the measurements, improves the overall stability, and minimizes the risk of damage. Consequently, our measurements reflect an average result where both strongly coupled signals (plasmonic system at the metasurface region with plasmon hot spots) and weakly coupled signals (plasmonic system at the metasurface region without plasmonic hot spots) contribute. Despite the existence of dense plasmonic hot spots in our system, the large laser beam sizes pose a challenge in differentiating the lifetimes between strong and ultra-strong couplings. It is noteworthy that the spatial resolution of our dark-field scattering measurements is about 500 nm . This high resolution allows us to precisely identify the position of plasmonic hotspots, facilitating the distinction between strong and ultra-strong couplings.

We thank the reviewer for the valuable suggestion. In our future research, we will aim to reduce the size of the laser beams while maintaining precise alignment between the pump and probe beams to ensure that lifetime measurements can clearly and accurately distinguish between strong and ultrastrong couplings.

Fig. R2| Ultrafast dynamics of the exciton-plasmon coupling in the WS_2 -Au plasmonic system in strong (SC) and ultrastrong (USC) coupling regimes. Time evolution of the $\Delta T/T$ signal for the WS_2 monolayer, pure Au metasurface, upper polariton branch in the strong and ultrastrong coupled WS_2 -Au system with pump-probe delay time of -0.5 to 7 ps (a) and -0.2 to 2 ps (b). SC: strong coupling. USC: ultrastrong coupling at -2% compressive strain.

Comment #3:

How to fit the data in Fig 2.a and b to obtain the coupling strength in Fig 2c? A detailed description is necessary, and the fitting process requires Au thickness but this parameter does not appear in equations 10 and 12.

Response:

We thank the reviewer for drawing our attention to the detailed description of the fitting process to obtain the coupling strength in Fig. 2c. In the theoretical fitting process, we first extract the characteristic polariton energies (i.e., lower (ω_-) and upper (ω_+) polaritons) from the scattering peaks in the spectra in Fig. 2a,b. We then calculate the coupling strength ($g = 165.9$ meV at 0 strain and $g = 240.4$ meV at -2% strain) by fitting the dark-field spectra in Fig. 2a,b to a coupled oscillator model³ where each scattering spectrum is calculated by $S(\omega) \propto \omega^4 \left| \frac{\omega_{ex}^2 - \omega^2 - i\gamma_{ex}\omega}{(\omega_{pl}^2 - \omega^2 + i\gamma_{pl}\omega)(\omega_{ex}^2 - \omega^2 + i\gamma_{ex}\omega) - \omega^2 g^2} \right|^2$. Here, only g is the unknown parameter, and ω_{ex} (known value) and ω_{pl} (calculated from the extracted ω_- and ω_+ as $\omega_{pl} = \omega_+ \omega_- / \omega_{ex}$) are the energies of the WS₂ excitons and the plasmonic mode, respectively, and $\gamma_{pl} = 380$ meV and $\gamma_{ex} = 45$ meV are the damping losses of plasmonic resonance and exciton emission, respectively. Third, we fit the polariton energies (ω_+ and ω_- extracted from the scattering peaks) as eigenvalues of the full Hopfield Hamiltonian^{4,5} which yields

$$(\omega^2 - \omega_{ex}^2)(\omega^2 - \omega_{pl}^2) - 4g^2\omega^2 = 0. \quad (\text{R1})$$

Note that, we do not need to use Au thickness parameters in the theoretical fitting process. In Fig. 2c of the original manuscript, the x -axis is the Au thickness showing that different ω_{pl} are introduced by different Au thicknesses. Different Au thicknesses are used to obtain different plasmonic resonances ω_{pl} and thus different detunings ($= \omega_{pl} - \omega_{ex}$, which is the difference between the plasmonic resonance and the exciton energy) to obtain anti-crossing. In fact, different detunings can be obtained in conventional plasmonic metasurfaces by using various techniques, for example, different detunings can be obtained in the well-known gold bowtie nanoantennas by introducing different metal film thicknesses (as in our case), different plasmonic gaps, or different bowtie side lengths, etc.

Changes introduced in the revised manuscript:

- 1) In the revised main text, section ‘**Ultrastrong exciton-plasmon coupling and its tunability**’, the 2nd sentence of the 2nd paragraph has been changed to: ‘Herein, to quantify the coupling strength, we extract the polariton energies (guided by the blue and purple curves in Fig. 2a,b) from the scattering peaks in the spectra and obtain the vacuum Rabi frequency by fitting the spectrum to a coupled oscillator model³ (See Section 4 in the SI for details).’

Comment #4:

Despite the explanation given in Fig. 3a, the mechanism for strain tuning the ultrastrong is not described. The compressive and tensile strain deviate the plasmonic resonance wavelength from the exciton wavelength in WS₂, but the compressive strain leads to the

coupling strength enhancement, which is opposite of the tensile strain. Please explain in detail the mechanism of coupling strength change under different compressive and tensile strain in Fig 3c.

Response:

The strain contributes to the strong/ultrastrong coupling process in two primary ways: first, it shifts the plasmonic resonance energy ω_{pl} (Fig. S3 in the previous Supplementary Information), and second, it tunes the gap size and thus the local field enhancement where a smaller plasmonic gap (which is kept larger than critical size for quantum tunneling, i.e. ~ 0.3 nm for the air gap in our plasmonic system) results in a larger local field enhancement. Specifically, the plasmonic response shifts to lower/higher energy when compressive/tensile strain is applied, and the local field enhancement is largely enhanced/reduced due to the corresponding reduced/enlarged gap size, and both effects lead to significant changes of the coupling strength. Regarding the referee's comment on the different change in coupling strength enhancement by compressive and tensile strains, it's important to note that the local field enhancement (E , where $g \propto E$) is enhanced/reduced by the reduced/enlarged gap size^{6,7} when compressive/tensile strain is applied. This explains why compressive strain increases the strength of the coupling, while tensile strain reduces it.

Changes introduced in the revised manuscript:

- 1) In the revised main text, we have added 'The plasmonic response shifts to lower/higher energy when compressive/tensile strain is applied, and the local field enhancement is significantly enhanced/reduced owing to the reduced/enlarged gap size, and both effects change the coupling strength.' to the caption of Fig. 3b.

Comment #5:

In Fig. S1, the schematics should be numbered and organized so that the reader can understand them. In addition, TPU, PDMS and PEI should be clearly annotated, Otherwise, it's easy to confuse the reader.

Response and changes introduced in the revised manuscript:

We thank the referee for this suggestion. Accordingly, we have added arrows to show the fabrication process in Fig. S1 in the revised Supplementary Information. PEI (poly(ethyleneimine)), TPU (thermoplastic polyurethane) and PDMS (poly(dimethylsiloxane)) and are clearly annotated in the corresponding caption, as shown in Fig. R3.

Fig. R3| Schematic illustration of the fabrication process of the WS₂ monolayer coupled multi-singular plasmonic metasurface. A thin gold film was deposited on a PEI polymer substrate (125 μm thick, smooth on both sides) using an electron beam evaporator at a deposition rate of 2 Ås⁻¹. During the evaporation, the temperature of the vacuum chamber was kept below 60 °C throughout the whole process, to prevent the thermal expansion or deformation of the PEI film due to built-in stress, which would cause wrinkles or defects in the gold film. For cold-etching, the first stretch was conducted by stretching the gold/PEI film in the x-direction. The second 2D stretch was conducted by re-stretching the as-fabricated film in the y-direction. After the cold-etching, the resulting gold nanopattern was transferred to a flexible TPU polymer substrate (2 mm thick) using a dry peel-off method. WS₂ monolayers were mechanically exfoliated from the purchased commercial bulk WS₂ crystals using a PDMS tape and transferred to the gold nanopatterns (on TPU) using a dry transfer method (optical microscopy images of the plasmonic system are shown in Fig. S2). This cold-etching technique is easy to implement, time-saving, low-cost, and large-scale self-assembly. The above sample preparation process demonstrates that cold-etching technology is easy to implement, time-saving, low-cost, and large-scale self-assembly. PEI: poly(ethyleneimine) polymer; Au: gold; TPU: thermoplastic polyurethane polymer; PDMS: poly(dimethylsiloxane) polymer.

Comment #6:

The ultrastrong coupling maybe led to occur Bose-Einstein condensation (BEC) phenomenon at low temperature, Is it possible that doing experiments at low temperature?

Response:

We thank the referee for this encouraging comment. Below are the dark-field scattering spectra of the monolayer WS₂ on a 20 nm thick plasmonic metasurface measured at different temperatures (Fig. R4, which is Fig. S16 in the revised Supplementary Information). We can see that the exciton resonance blue-shifts on cooling, which is in agreement with the standard semiconductor behavior where the temperature dependence of the semiconductor bandgap (E_g) is conveniently described by the O'Donnell model⁸

$$E_g(T) = E_g(0) - S\langle\hbar\omega\rangle\{\coth[\langle\hbar\omega\rangle/2k_B T] - 1\} \quad (R2)$$

where $E_g(0) = 2.07$ eV is the bandgap at 0 K, $S = 1.78$ is the electron-phonon coupling strength, and $\langle \hbar\omega \rangle = 25$ meV is the average phonon energy, respectively.

Since exciton-polariton condensation exhibits most of the features of Bose-Einstein condensation^{9,10}, and such exciting ideas deserve systematic investigation. We will explore the possibility of building polariton lasers based on an atomically thin TMDs in future work.

Fig. R4| Dark-field scattering spectra of the monolayer WS₂ on a 20 nm thick plasmonic metasurface at different temperatures. The exciton resonance blue-shifts on cooling, which is in agreement with the standard semiconductor behavior where the temperature dependence of the semiconductor bandgap (E_g) is conveniently described by the O'Donnell model⁸ $E_g(T) = E_g(0) - S\langle \hbar\omega \rangle \{ \coth[\langle \hbar\omega \rangle / 2k_B T] - 1 \}$. $E_g(0) = 2.07$ eV is the bandgap at 0 K, $S = 1.78$ is the electron-phonon coupling strength, and $\langle \hbar\omega \rangle = 25$ meV is the average phonon energy, respectively.

Changes introduced in the revised version:

- 1) In the revised Supplementary Information, we have added Fig. R4 as Fig. S16.

Comment #7:

The equipment and method of the strain application should be described in the supplementary material, to help readers to understand the process. Furthermore, the schematic of dark field scattering spectra under strain should be shown in the supplementary material to the reader understand this test method.

Response and changes introduced in the revised manuscript:

We thank the reviewer very much for the two suggestions. Following this comment, we have added the experimental setup for applying uniaxial strain (Fig. R5, which is Fig. S8 in the revised Supplementary Information). During experimental measurements, the positive strain (via upward bending the sample as shown in Fig. R5a) is tensile on the ‘outside’ surface and the negative strain (via downward bending the sample as shown in Fig. R5b) is compressive on the ‘inside’ surface. We have added the schematic of the dark-field scattering measurement setup in the revised Supplementary Information (Fig. R6, which is Fig. S9 in the revised Supplementary Information).

Fig. R5| Experimental setup to apply uniaxial strain by bending the flexible substrate. a. The positive strain (upward bending) is tensile on the ‘outside’ surface. **b.** The negative strain (downward bending) is compressive on the ‘inside’ surface.

Fig. R6| Schematic of the dark-field scattering measurement setup.

Referee #2:

Light-matter interaction has been an important topic for more than a hundred year. Recently, in the quantum regime, ultrastrong coupling is highly demanded. In this work, the authors fabricated a metasurface with many nanometre-sized plasmonic hotspots to which WS₂ excitons coherently interact, achieving the ultrastrong exciton-plasmon coupling. This work is interesting. Before making the final recommendation, I have a couple of questions.

Response:

We greatly appreciate your deep and thorough review of our manuscript. Based on your constructive comments and suggestions, we have revised and improved our manuscript. In the following, we provide a detailed point-by-point response to your comments and questions.

Comment #1:

Usually, ultrastrong coupling can overcome the disturbance caused by the linewidth. Here, the linewidth is still very broad. Please comment on this issue and its influence on the significance of ultrastrong coupling, especially its possible application in quantum optics.

Response:

We thank the reviewer for this constructive suggestion. The effects of the linewidth are described as follows:

1) The influence of the linewidth on the significance of ultrastrong coupling.

It should be noted that ultrastrong coupling (USC) is not strong coupling (SC) with larger couplings. The definitions of the weak, strong and ultrastrong coupling regimes compare the light-matter coupling strength g with different parameters. Whether the coupling is strong or weak depends on whether g is greater than the losses in the system ($(\gamma_{pl} + \gamma_{ex})/4$), i.e., if $g > (\gamma_{pl} + \gamma_{ex})/4$, the coupling is strong, whereas if $g < (\gamma_{pl} + \gamma_{ex})/4$, the coupling is weak. The definition of USC is irrelevant to the losses, but instead depends on the energies of the system (i.e. the excitation energy ω_{ex}). USC requires that the normalized coupling strength, defined as $\eta = g/\omega_{ex}$ is greater than 0.1. It is therefore possible for a system to be in the USC regime but not in the SC regime if the losses are sufficiently large¹¹.

We would like to point out here that our system works not only in the USC regime, but also in the SC regime, since $g > (\gamma_{pl} + \gamma_{ex})/4$ is also satisfied, where the lowest g in our system is 165.9 meV with the damping losses of plasmonic resonance and exciton emission being $\gamma_{pl} = 380$ meV and $\gamma_{ex} = 45$ meV, respectively (see the last sentence of the 1st paragraph of Section ‘**Ultrastrong exciton-plasmon coupling and its tunability**’ of the original manuscript).

- 2) The influence of the linewidth on the possible applications of our plasmonic system in quantum optics.

In response to this comment, we have carefully analyzed the effect of the linewidth on possible applications of our plasmonic system in quantum optics and found that the implications of our findings of USC in WS₂ multilayers for applications in quantum optics are not straightforward and would require a lengthy explanation (as below). Therefore, we have removed this motivation from the revised manuscript. We plan to investigate in depth the possible applications in quantum optics in a future work and we thank the referee for this pertinent suggestion.

To address the interesting point raised by the reviewer, we discuss below the influence of linewidth on the potential applications of our plasmonic system in quantum information processing (QIP).

Using an input-output process relevant to low-Q optical cavities¹²⁻¹⁴, we show that our plasmonic system can serve as a promising candidate setup to meet the requirements of QIP schemes. The key step is to design a scheme for entangling two atoms confined respectively in two spatially separated low-Q plasmonic cavities. We first present an analytical expression for the reflection rate of the input-output process for a plasmonic cavity interacting with a trapped two-level atom^{15,16}

$$r(\omega) = \frac{a_{out}(t)}{a_{in}(t)} = \frac{[i(\omega_{pl}-\omega) - \frac{\gamma_{pl}}{2}][i(\omega_{ex}-\omega) + \frac{\gamma_{ex}}{2}] + g^2}{[i(\omega_{pl}-\omega) + \frac{\gamma_{pl}}{2}][i(\omega_{ex}-\omega) + \frac{\gamma_{ex}}{2}] + g^2} \quad (R3)$$

For the case $g = 0$, i.e., the TMDs are uncoupled from the cavity, Eq. (R3) gives

$$r_0(\omega) = \frac{i(\omega_{pl}-\omega) - \frac{\gamma_{pl}}{2}}{i(\omega_{pl}-\omega) + \frac{\gamma_{pl}}{2}} \quad (R4)$$

For the resonant condition $\omega = \omega_{pl} = \omega_{ex}$, since the input photon (ω) is in resonance with the bare plasmonic cavity ($\omega_{pl} = \omega_{ex}$), we have

$$r(\omega_{pl}) = \frac{g^2 - \gamma_{pl}\gamma_{ex}/4}{g^2 + \gamma_{pl}\gamma_{ex}/4} \quad (R5a)$$

$$r_0(\omega) = -1 \quad (R5b)$$

If g is much larger than γ_{pl} and γ_{ex} , $r(\omega_{pl})$ is approximately equal to 1, meaning that the input photon remains unchanged when it goes out of the system. When we set the coupling strength $g = 2.66\sqrt{\gamma_{pl}\gamma_{ex}}$ (corresponding to the case of the WS₂ quadrilayer synthesized with the multi-singular metasurface with $[g, \gamma_{pl}, \gamma_{ex}] = [328.5, 380, 40]$ meV), the reflection coefficient $r(\omega_{pl}) = 0.932$ is close to 1, which allows an almost unchanged reflection without the strict requirement of high-Q cavities. In contrast, in the case where the plasmonic cavity mode (ω_{pl}) is far detuned with respect to the confined atom (ω_{ex}), we can equivalently consider the model as the photon resonantly

integrating with a bare cavity, yielding $r_0(\omega_{pl}) = -1$ where the large detuning between the cavity mode and the atom means $g = 0$. Thus, consider the coupled system prepared in the state $|-1\rangle|L\rangle$ under the resonant condition: the single photon enters and leaks out of the system without being absorbed by the cavity mode, i.e., $|\Psi'_{out}\rangle = r(\omega_{pl})|-1\rangle|L\rangle \simeq |-1\rangle|L\rangle$; otherwise, $r_0(\omega) = -1$ (Eq. (R5b)) means $e^{i\phi_0} = e^{i\pi}$ with a phase shift of π . Then, the single photon pulse travels through two plasmonic systems sequentially and a successful detection of the output photon pulse leads to entanglement between the two plasmonic systems for QIP implementation. Therefore, the input-output process of single photon pulses, which is easily scalable to the entanglement of many separate WS₂ synthesized multi-singular plasmonic metasurface, will be useful for the construction of a working quantum network.

Changes introduced in the revised manuscript:

- 1) In the revised main text, we changed the last sentence of the Section ‘**Discussion**’ to ‘Our work could also lead to new insights in fundamental science and potential applications in the fields of nonlinear nanophotonics and analytical chemistry, among others.’.

Comment #2:

From Fig. 2a-b to Fig. 2c, how did the authors extract the resonant frequencies of the two branches? Please show the details, since the two original resonances have significantly different linewidth.

Response and changes introduced in the revised manuscript:

Experimentally, we extract the polariton energies (i.e., lower (ω_-) and upper (ω_+) polaritons) from the scattering peaks in the spectra in Fig. 2a,b. In the theoretical fit process (Fig. 2c), we first calculate the coupling strength ($g = 165.9$ meV at 0 strain and $g = 240.4$ meV at -2% strain) using the above extracted experimental ω_- and ω_+ by fitting the dark-field spectra in Fig. 2a,b to a coupled oscillator model³ where each scattering spectrum is calculated by $S(\omega) \propto \omega^4 \left| \frac{\omega_{ex}^2 - \omega^2 - i\gamma_{ex}\omega}{(\omega_{pl}^2 - \omega^2 + i\gamma_{pl}\omega)(\omega_{ex}^2 - \omega^2 + i\gamma_{ex}\omega) - \omega^2 g^2} \right|^2$. Here, only g is an unknown parameter and can be calculated, and ω_{ex} (known value) and ω_{pl} (calculated from the extracted ω_- and ω_+ as $\omega_{pl} = \omega_+ \omega_- / \omega_{ex}$) are the energies of the WS₂ excitons and the plasmonic mode, respectively, and $\gamma_{pl} = 380$ meV and $\gamma_{ex} = 45$ meV are the damping losses of plasmonic resonance and exciton emission, respectively. We then fit the experimental polariton energies (extracted ω_- and ω_+) as eigenvalues of the full Hopfield Hamiltonian^{4,5} which yields $(\omega^2 - \omega_{ex}^2)(\omega^2 - \omega_{pl}^2) - 4g^2\omega^2 = 0$. Referee #1 raised a similar question in his/her 3rd comment. Please also refer to our response to that comment.

Comment #3:

For each measurement, the light spot would cover a wide range of hotspots, i.e., the LSPR peak here is from an ensemble of nanogaps. Did the author measure the fluctuation of the LSPR peak of each sample? If so, at least one example should be shown, which is the scattering spectra from different positions of a single sample. Also, a standard error should be given. Only in this way, the potential readers could replicate all the experiments and data.

Response and changes introduced in the revised manuscript:

This comment is similar to referee #1's 1st comment, we refer the referee to our previous response.

In response to the suggestion to add error bars, we have incorporated them into the normalized coupling strength and the ground-state energy modification in Fig. 3c,d in the manuscript, and we have explained that the error bars in Fig. 3c,d result from the variation in dark-field scattering from multiple batches of samples and are extracted from the standard error of the fit.

Comment #4:

Please comment on the fact in Fig. 2d: the splitting of SHG spectra is not as large as that of the scattering spectra.

Response and changes introduced in the revised manuscript:

We thank the referee for this suggestion and agree that there is a need to comment on the deviation of the splitting of the SHG spectra from the linear scattering spectra. Following this comment, we have included the following statement in the caption of Fig. 2d in the revised manuscript: ‘Note that the deviation between SHG and dark-field scattering splittings most likely arises from measurement errors in the strength of the SHG signal at the spectral edge of the photon detector.’.

Comment #5:

Fig. S15 is not cited in the main text. Please further comment on the mechanism why the upper branch is not shown in PL spectra.

Response and changes introduced in the revised manuscript:

We thank the referee for this valuable suggestion. Strong plasmon-exciton coupling has been mostly studied using dark-field scattering and reflection/transmission spectra, and only a few experiments have measured PL spectra. For single quantum dots coupled to plasmonic nanostructures, an obvious splitting of the PL spectral has been observed, as there is no background PL from the excitons uncoupled to the plasmonic structures¹⁷. In sharp contrast, in J aggregates, dye molecules and two-dimensional transition metal dichalcogenides, PL from only the lower polariton state or overlapped PL spectral of the

lower and upper polariton states has been observed¹⁸⁻²⁷. In these strongly coupled systems, part of the detected PL (PL_{detected}) is emitted by plasmon-exciton hybrid modes, which can modulate the PL spectra^{28,29}, and the rest of the detected PL signal is attributed to the background PL from the uncoupled excitons ($PL_{\text{uncoupled}}$). Extracting the strong coupling information from the PL spectra thus becomes more difficult. Separating the PL emission due to the radiation of plasmon-exciton hybrid modes from that due to uncoupled excitons (i.e., $PLE = (PL_{\text{detected}} - PL_{\text{uncoupled}})/PL_{\text{uncoupled}}$) can help to reveal the strong coupling induced spectral features of PL.

Following the referee's suggestion, here we also compare the normalized PL (i.e., $PLE = (PL_{\text{WS}_2 \text{ on metasurface}} - PL_{\text{WS}_2 \text{ on polymer}})/PL_{\text{WS}_2 \text{ on polymer}}$) to the scattering in the SC (upper, strain=0) and USC (lower, strain=-2%) regime, as shown in Fig. R7 (Fig. S19b in the revised Supplementary Information). The normalized PL spectra show almost the same profiles as the corresponding scattering spectra, indicating that the splitting in the PL is slightly narrower than the splitting in the scattering spectra, as has been verified by Wersäll et al.²⁰. We attribute this to the absorption and scattering paths associated with the PL and dark-field scattering processes, respectively.

Fig. R7| Normalized PL intensity ($PLE = (PL_{\text{WS}_2 \text{ on metasurface}} - PL_{\text{WS}_2 \text{ on polymer}})/PL_{\text{WS}_2 \text{ on polymer}}$) compared to scattering in the SC (upper, strain=0) and USC (lower, strain=-2%) regimes, respectively. The splitting in the PL spectra is narrower than that in the scattering cross sections, as has been also observed in previous SC experiments²⁰. We attribute this to the absorption and scattering paths associated with the PL and dark-field scattering processes, respectively.

Comment #6:

Grammar and format.

Response and changes introduced in the revised manuscript:

We thank the referee for this comment. We have carefully revised the article statements to correct grammatical errors.

Referee #3:

Strong coupling between 2D exciton and surface plasmon has been intensively studied in the past few years and has resulted in a number of published papers. In this work, the authors report on the observation of ultrastrong coupling of surface plasmon and exciton in two-dimensional van der Waals semiconductor (monolayer and multilayer tungsten disulfide). This coupling is realized by formation of random plasmonic “hot gaps” via etching of a thin gold film deposited on a flexible polymer substrate. The main differences from published works are the observation of large Rabi splitting (300 – 450 meV) and mechanical tunability due to the use of a flexible substrate. However, there are two obvious drawbacks: (1) Their approach is stochastic in nature and difficult to have real applications (e.g., lasing, nonlinear optics claimed by the authors), (2) ultrastrong coupling (large Rabi splitting) in a plasmonic platform has previously been reported. My evaluation is to enhance the technical contents and publish the revised paper in a regular journal, since this work does have the novelty and urgency to warrant publication in Nature Communications.

Response and changes introduced in the revised manuscript:

We would like to thank the reviewer for his/her efforts in reviewing our work.

(1) Regarding the comment on the method is stochastic in nature, we have used a theoretical strain model to guide our design and fabrication of the multi-singular plasmonic metasurface, such that the plasmonic gap sizes and the global uniformity are well-controlled. As demonstrated by further experiments (please refer to our replies to comment #1 of the first referee), the fabrication of multi-singular metasurfaces is quite reliable, stable, and repeatable at the macroscopic scale. The multi-singular plasmonic metasurfaces exhibit randomness only in the distribution and orientation of the singularities within the nanoscale area. They still possess global uniformity, implying that the distribution of singularities across the entire patterned gold film is globally uniform with macroscale reliability, stability, and repeatability (refer to Fig. 1e in the manuscript and Fig. R8 for more details). In the revised manuscript and Supplementary Information, we have provided specific explanations to clarify that:

a) The physical morphology and corresponding optical properties of the multi-singular plasmonic metasurface are stable and reliable, highlighting its most attractive property for USC: large local field enhancements over a large surface area.

b) The cold-etching technique itself is easy to implement, time-efficient, cost-effective, and conducive to large-scale self-assembly.

Regarding the concern about the method's applicability to real-world scenarios, the global uniformity promises important practical applications of this multi-singular plasmonic metasurface, such as plasmonic enhanced harmonic generation³⁰ (see Fig. R9 which is Fig. S18 in the revised Supplementary Information), surface-enhanced Raman scattering³¹ (Fig. R10) and biosensing³² (Fig. R11), among others.

(2) Regarding the comment about previous reports of ultrastrong coupling (large Rabi splitting) in plasmonic platforms, firstly, we would like to highlight that, although ultrastrong coupling (USC) in plasmonic platforms has been reported³⁴⁻³⁷, previous works either rely on bulk materials or operate at cryogenic temperature. Our work is the first experimental demonstration of USC using 2D materials. Furthermore, our reported Rabi splitting of 634.7 meV exceeds all previously documented values at optical frequencies even when compared with materials that are 100 times thicker, to the best of our knowledge. We highlight that such a large Rabi splitting is quite difficult, if not impossible, to achieve with conventional plasmonic platforms, underscoring the novelty of our proposed multi-singular random metasurface. Moreover, utilizing 2D materials to achieve USC offers several advantages over bulky materials. Firstly, the resulting device is more compact and easier to scale up. Secondly, 2D materials exhibit exotic excitonic and nonlinear properties. For example, transition-metal dichalcogenides (TMDs) have direct bandgaps at visible frequencies, large exciton binding energies, pronounced resonance strengths, and narrow linewidths even beyond the room temperature^{38,39}. Exploring USC with 2D materials, especially TMD monolayers, will open up great opportunities for various technologies, including harmonic generation in solids, surface-enhanced Raman scattering, and broadband optical tunability, demonstrating the urgency of our approach. Last but not least, our work provides a feasible route to reconfigurably tune the normalized coupling strength from 0.075 to 0.164 by strain/stress engineering. This is also the first demonstration of tunable USC using 2D materials.

Finally, we appreciate the suggestions to “*enhance the technical contents*” in the revised submission. The major changes in the revised version include:

- 1) In the revised main text, we have added ‘Note that, despite of the random distribution and orientation of the singularities, the fabricated multi-singular metasurface maintains a ‘global uniformity’, which refers to the uniform distribution of singularities within the laser beam area or across the entire patterned gold film. This global uniformity can be precisely controlled during the cold-etching process (see Fig. S7 in SI for details).’ to the caption of Fig. 1e.
- 2) In the revised main text, we have added ‘If we assume that the experimental gap size follows a normal distribution, $L_{\text{gap}} \sim N(\mu, \sigma^2)$, where $\mu = \bar{L}_{\text{gap}}$ and the variance σ^2 can be obtained from experimental measurements. The number of nanometre-sized gaps (L_{nano}) can be obtained from the normal distribution by determining the area satisfying $L_{\text{gap}} - \mu < L_{\text{nano}}$.’ to the end of the ‘**Strain mode**’ in the Methods section. The purpose of adding this sentence is to indicate that the strain model serves as a complete physical guide for determining both the average size and density of the nanometre-sized gap.
- 3) In the revised Supplementary Information, we have added ‘The above sample preparation process demonstrates that cold-etching technology is easy to implement, time-saving, low-cost, and large-scale self-assembly.’ to the caption of Fig. S1.

- 4) In the revised Supplementary Information, we have added Fig. R8 as Fig. S10 to show the global uniformity in detail.
- 5) Other changes are marked in the manuscript.

Fig. R8| Dark-field scattering spectra measured at 10 different random positions of a 20 nm thick Au metasurface **a**, without and **b**, with monolayer WS₂ on top. The relative standard deviation values at the resonance wavelength/bandwidth of the bare plasmons and the Rabi splitting are 6.2%/8.2% and 7.5%, respectively, indicating the stability and repeatability of our random multi-singular metasurface. **c**, Dark-field scattering spectra at 6 different random positions of a 20 nm thick Au metasurface. Black lines correspond to the signal of a newly prepared sample, blue lines correspond to the signal repeated 12 months after the samples were fabricated at the same sample positions. No significant difference in the scattering spectra when the measurements were repeated 12 months after the samples were fabricated, further demonstrating the stability and robustness of the plasmonic metasurfaces.

Fig. R9| Tunable resonant polariton-enhanced second-order nonlinearity in WS₂ monolayer. **a**, Schematic energy diagram describing the polariton-enhanced second harmonic generation (SHG) in the WS₂ monolayer arising from the two-photon resonances in the vicinity of the two hybridized polaritonic states between 2D excitons and surface plasmons. **b**, Resonant polariton-enhanced second harmonic (SH) signal intensity (I_{SH}) from WS₂ as a function of the SHG energy. The peak of the I_{SH} in the WS₂ monolayer on polymer originates from the two-photon exciton resonance enhanced SHG. The peaks and splittings of the I_{SH} in the WS₂ monolayer coupled to a multi-singular metasurface are due to the two-photon resonances provided by the hybridized polaritonic states. Compared to the SC, USC provides broader and higher SHG enhancement. **c**, Polarized polariton-enhanced SHG from the WS₂ monolayer. A typical 6-fold symmetric petal SHG pattern is well fitted by $I_{SH} = C \cos^2 3\theta$ showing the isotropic property of the multi-singular plasmonic metasurface, where C is the maximum SHG intensity, and θ is the rotation angle of the sample. The intensity of the SHG emission from the WS₂ on PDMS is amplified by a factor of three for a clear view. The polariton-enhanced SHG from the WS₂ monolayer is around 15 times stronger than that on PDMS. The polarization-independent polariton-enhanced SHG is promising because it eliminates the need to align the crystal orientation of the WS₂ for effective polariton-enhanced nonlinearity, in contrast to the polarization-dependent SHG in the WS₂ coupled bowtie antenna system where maximum enhancement occurs only when the pump field is polarized along the longitudinal symmetry axis. **d**, SHG intensity under different uniaxial strains. Error bars are standard errors from multiple samples. In **c** and **d**, the laser wavelength is fixed at 800 nm. Note that the isotropic metasurface has very uniformly distributed nanometer-sized plasmonic gaps and the laser beam has a fairly large spot size, so that each SHG measurement sees an average effect, resulting in particularly reproducible SHG intensity under different uniaxial strains.

Fig. R10| **a**, Raman spectrum of the graphene layer on silicon (g-Si) and on Au multi-singular metasurface (g-Au) substrates at different Au thickness. The green vertical strips highlight the G and G' graphene Raman bands. The raman single from the graphene on silicon is magnified 10 times for better visibility. **b**, Histogram of the surface-enhanced Raman scattering (SERS) enhancement factor of the Au metasurface at different Au thickness (t_{Au}).

Fig. R11| **Probing R6G molecular fingerprints using the Au multi-singular metasurface. a**, Surface-enhanced Raman scattering spectra of R6G molecule at concentrations ranging from 1 pM to 10 μM, with the yellow vertical strip highlighting the typical band of the R6G molecule **b**, Peak intensities at 1310 cm⁻¹ band (typical band of the R6G molecule) as a function of R6G concentration.

Comment #1:

The imaginary part (linewidth) evolution in the ultracoupling region should be analyzed and presented, especially for a coupling system with large difference in linewidth (380 meV vs. 45 meV).

Response and changes introduced in the revised manuscript:

We appreciate the constructive comment from the referee. In response, we have included the linewidth as a function of the coupling strength at zero detuning under two different strains (Fig. R12, which is Fig. S23 in the revised Supplementary Information). Notably, an asymmetry in the polaritonic linewidths is observed, where the decay rate of the upper panel consistently exceeds that of the lower panel, with the degree of asymmetry increasing

with the coupling strength. This phenomenon aligns with observations in other materials⁴⁰⁻⁴², offering promising implications. Specifically, the ultrastrong coupling between lossy plasmons and lossy TMDs may lead to a very narrow lower panel polariton, which holds potential for various applications, such as sensing.

Fig. R12| Linewidth of the lower (LPB) and upper (UPB) polariton branches as a function of the normalized coupling strength (g/ω_{ex}) at zero detuning under 0 (black) and -2% (blue) strains. ($(\gamma_{pl} + \gamma_{ex})/2$ shows the average linewidth of the coupled plasmonic system. The polaritonic linewidths are asymmetric, where the decay rate of the upper panel is always greater than that of the lower panel, and the larger the coupling strength the greater the asymmetry. This asymmetry is very interesting and suggests that the ultrastrong coupling between lossy plasmons and lossy TMDs can lead to a very narrow lower panel polariton, which can be used for some other interesting applications, such as sensing.

Comment #2:

It is necessary to provide the detailed information about coupling of exciton and plasmon in the case of multilayer WS₂ flake, since intra and interlayer excitons have different dipole orientations and different coupling strengths with plasmon in “hot spots”.

Response and changes introduced in the revised manuscript:

We thank the referee very much for this constructive comment. Indeed, discussing the dependence of the in-plane transition dipole moment (μ_{xy}) of WS₂ on the layer number provides valuable insight into understanding the effect of layer number on the coupling strength ($g = \sqrt{N}\mu_{xy}E_{xy}$, where the subscript xy denotes the in-plane direction, and we focus solely on in-plane coupling due to the negligible contribution of the out-of-plane plasmonic field component, E_z).

In response, we have investigated the evolution of μ_{xy} as a function of the layer number by leveraging the absorption per layer (A) at the exciton transition. This analysis is based on the relationship $\mu_{xy} \propto \sqrt{A}$ and the known value of $\mu_{xy} = 56$ Debye (D) for monolayer WS₂^{27,43}. The absorption spectra of the multilayer WS₂ in Fig. R13a demonstrate that the

in-plane dipole moments gradually decrease from 56 D for monolayer to 39.6 D for trilayer and 37.5 D for quadrilayer cases, respectively.

To fully understand the effects of the layer number on the coupling strength, we present a comparison between the measured and estimated coupling strength ($g = \sqrt{N}\mu_{xy}E_{xy}$, where E_{xy} is the in-plane plasmonic electric field averaged over the unit cell in the FDTD simulations at half height of the WS₂ flakes) as a function of the layer number in Fig. R13b (Fig. S21 in the revised Supplementary Information). It can be seen that the experimental results are in good agreement with the theoretical estimates, indicating that consideration of in-plane coupling suffices due to the screened normal field.

Fig. R13| a, Normal incidence absorption of the excitons in monolayer, trilayer and quadrilayer WS₂. The evolution of the in-plane transition dipole moment (μ_{xy}) of the WS₂ as a function of the layer number is determined using the absorption per layer (A) at the exciton transition, based on the relationship $\mu_{xy} \propto \sqrt{A}$ and the known value of $\mu_{xy} = 56$ Debye (D) for monolayer^{27,43}. The absorption of the multilayer WS₂ reveal that μ_{xy} gradually decrease from 56 D for monolayer to 39.6 D for trilayer and 37.5 D for quadrilayer cases, respectively. **b**, Measured (orange and purple) and estimated ($g = \sqrt{N}\mu_{xy}E_{xy}$, where E_{xy} is the in-plane plasmonic electric field averaged over the unit cell in the FDTD simulations at half height of the WS₂ flakes, black dashed lines) normalized coupling strength (g/ω_{ex}) as a function of the WS₂ layers under 0 (orange) and -2% compressive strain (purple). The experimental results are in good agreement with the theoretical estimates, showing that the in-plane coupling is only needs to be considered due to the screened normal field. Error bars are derived from the variation in dark-field scattering from different batches of samples and are extracted from the standard error of the fit.

References

1. Shabahang, S. *et al.* Controlled fragmentation of multimaterial fibres and films via polymer cold-drawing. *Nature* **534**, 529-533 (2016).
2. Wu, T. *et al.* Ultrawideband surface enhanced Raman scattering in hybrid graphene fragmented-gold substrates via cold-etching. *Adv. Opt. Mater.* **7**, 1900905 (2019).
3. Leng, H., Szychowski, B., Daniel, M.-C. & Pelton, M. Strong coupling and induced transparency at room temperature with single quantum dots and gap plasmons. *Nat. Commun.* **9**, 4012 (2018).
4. Ciuti, C. & Carusotto, I. Input-output theory of cavities in the ultrastrong coupling regime: The case of time-independent cavity parameters. *Phys. Rev. A* **74**, 033811 (2006).
5. Hopfield, J. Theory of the contribution of excitons to the complex dielectric constant of crystals. *Phys. Rev.* **112**, 1555 (1958).
6. Zhu, W. & Crozier, K. Quantum mechanical limit to plasmonic enhancement as observed by surface-enhanced Raman scattering. *Nat. Commun.* **5**, 5228 (2014).
7. Simmons, J. G. Generalized formula for the electric tunnel effect between similar electrodes separated by a thin insulating film. *J. Appl. Phys.* **34**, 1793-1803 (1963).
8. O'donnell, K. P. & Chen, X. Temperature dependence of semiconductor band gaps. *Appl. Phys. Lett.* **58**, 2924-2926 (1991).
9. Shan, H. *et al.* Second-Order Temporal Coherence of Polariton Lasers Based on an Atomically Thin Crystal in a Microcavity. *Phys. Rev. Lett.* **131**, 206901 (2023).
10. Byrnes, T., Kim, N. Y. & Yamamoto, Y. Exciton-polariton condensates. *Nat. Phys.* **10**, 803-813 (2014).
11. De Liberato, S. Virtual photons in the ground state of a dissipative system. *Nat. Commun.* **8**, 1465 (2017).
12. Duan, L.-M. & Kimble, H. Scalable photonic quantum computation through cavity-assisted interactions. *Phys. Rev. Lett.* **92**, 127902 (2004).
13. Cho, J. & Lee, H. Generation of atomic cluster states through the cavity input-output process. *Phys. Rev. Lett.* **95**, 160501 (2005).
14. Deng, Z., Zhang, X., Wei, H., Gao, K. & Feng, M. Implementation of a nonlocal N-qubit conditional phase gate by single-photon interference. *Phys. Rev. A* **76**, 044305 (2007).
15. An, J.-H., Feng, M. & Oh, C. Quantum-information processing with a single photon by an input-output process with respect to low-Q cavities. *Phys. Rev. A* **79**, 032303 (2009).
16. Hu, C., Young, A., O'brien, J., Munro, W. & Rarity, J. Giant optical Faraday rotation induced by a single-electron spin in a quantum dot: applications to entangling remote spins via a single photon. *Phys. Rev. B* **78**, 085307 (2008).
17. Leng, H. *et al.* Strong coupling and induced transparency at room temperature with single quantum dots and gap plasmons. *Nat. Commun.* **9**, 4012 (2018).
18. Rodriguez, S., Feist, J., Verschuuren, M., Vidal, F. G. & Rivas, J. Thermalization and cooling of plasmon-exciton polaritons: towards quantum condensation. *Phys. Rev. Lett.* **111**, 166802 (2013).
19. Bellessa, J., Bonnand, C., Plenet, J. & Mugnier, J. Strong coupling between surface plasmons and excitons in an organic semiconductor. *Phys. Rev. Lett.* **93**, 036404 (2004).
20. Wersall, M., Cuadra, J., Antosiewicz, T. J., Balci, S. & Shegai, T. Observation of mode splitting in photoluminescence of individual plasmonic nanoparticles strongly coupled to molecular excitons. *Nano Lett.* **17**, 551-558 (2017).
21. Wersäll, M. *et al.* Correlative dark-field and photoluminescence spectroscopy of individual plasmon-molecule hybrid nanostructures in a strong coupling regime. *ACS Photon.* **6**, 2570-2576 (2019).
22. Wang, S. *et al.* Coherent coupling of WS₂ monolayers with metallic photonic nanostructures at room temperature. *Nano Lett.* **16**, 4368-4374 (2016).
23. Kleemann, M.-E. *et al.* Strong-coupling of WSe₂ in ultra-compact plasmonic nanocavities at room temperature. *Nat. Commun.* **8**, 1296 (2017).
24. Qin, J. *et al.* Revealing strong plasmon-exciton coupling between nanogap resonators and two-dimensional semiconductors at ambient conditions. *Phys. Rev. Lett.* **124**, 063902 (2020).
25. Jiang, Y., Wang, H., Wen, S., Chen, H. & Deng, S. Resonance coupling in an individual gold nanorod-monolayer WS₂ heterostructure: photoluminescence enhancement with spectral broadening. *ACS Nano* **14**, 13841-13851 (2020).

26. Liu, L. *et al.* Strong plasmon–exciton interactions on nanoantenna array-monolayer WS₂ hybrid system. *Adv. Opt. Mater.* **8**, 1901002 (2020).
27. Liu, L. *et al.* Plasmon-induced thermal tuning of few-exciton strong coupling in 2D atomic crystals. *Optica* **8**, 1416-1423 (2021).
28. Zhao, L., Ming, T., Chen, H., Liang, Y. & Wang, J. Plasmon-induced modulation of the emission spectra of the fluorescent molecules near gold nanorods. *Nanoscale* **3**, 3849-3859 (2011).
29. Ringler, M. *et al.* Shaping emission spectra of fluorescent molecules with single plasmonic nanoresonators. *Phys. Rev. Lett.* **100**, 203002 (2008).
30. Leskova, T., Leyva-Lucero, M., Mendez, E., Maradudin, A. & Novikov, I. The surface enhanced second harmonic generation of light from a randomly rough metal surface in the Kretschmann geometry. *Opt. Commun.* **183**, 529-545 (2000).
31. Nishijima, Y., Khurgin, J. B., Rosa, L., Fujiwara, H. & Juodkazis, S. Randomization of gold nanobrick arrays: a tool for SERS enhancement. *Opt. Express* **21**, 13502-13514 (2013).
32. Lee, S. Y. *et al.* Spatial and spectral detection of protein monolayers with deterministic aperiodic arrays of metal nanoparticles. *PNAS* **107**, 12086-12090 (2010).
33. Schokker, A. H. & Koenderink, A. Lasing in quasi-periodic and aperiodic plasmon lattices. *Optica* **3**, 686-693 (2016).
34. Baranov, D. G. *et al.* Ultrastrong coupling between nanoparticle plasmons and cavity photons at ambient conditions. *Nat. Commun.* **11**, 2715 (2020).
35. Yoo, D. *et al.* Ultrastrong plasmon-phonon coupling via epsilon-near-zero nanocavities. *Nat. Photon.* **15**, 125-130 (2021).
36. Cacciola, A., Di Stefano, O., Stassi, R., Saija, R. & Savasta, S. Ultrastrong coupling of plasmons and excitons in a nanoshell. *ACS Nano* **8**, 11483-11492 (2014).
37. Mueller, N. S. *et al.* Deep strong light-matter coupling in plasmonic nanoparticle crystals. *Nature* **583**, 780-784 (2020).
38. Zhang, L., Gogna, R., Burg, W., Tutuc, E. & Deng, H. Photonic-crystal exciton-polaritons in monolayer semiconductors. *Nat. Commun.* **9**, 1-8 (2018).
39. Gu, J., Chakraborty, B., Khatoniar, M. & Menon, V. A room-temperature polariton light-emitting diode based on monolayer WS₂. *Nat. Nanotechnol.* **14**, 1024-1028 (2019).
40. Whittaker, D. *et al.* Motional narrowing in semiconductor microcavities. *Phys. Rev. Lett.* **77**, 4792 (1996).
41. Wang, W. *et al.* Interplay between strong coupling and radiative damping of excitons and surface plasmon polaritons in hybrid nanostructures. *ACS Nano* **8**, 1056-1064 (2014).
42. Wurdack, M. *et al.* Motional narrowing, ballistic transport, and trapping of room-temperature exciton polaritons in an atomically-thin semiconductor. *Nat. Commun.* **12**, 5366 (2021).
43. Sie, E. J. *et al.* Optical Stark effect in 2D semiconductors, *SPIE* **9835**, 129-137 (2016).

REVIEWERS' COMMENTS

Reviewer #1 (Remarks to the Author):

The authors have addressed all my concerns, it is now suitable for publish in NC.

Reviewer #2 (Remarks to the Author):

The authors have addressed the previous concerns and made notable improvements to the manuscript. While the observation of ultra-strong coupling between 2D material and plasmonic metasurface is impressive, it appears that the potential influence may not be as extensive as initially anticipated. The deterministic ultra-strong coupling between a single nanogap and 2D material remains unproven, as the observed Rabi splitting is the average value of an ensemble of detuned strong coupling systems. Additionally, while the wide linewidth does not impact the definition of ultra-strong coupling, it does limit the potential applications. Therefore, I believe this work could be published, but perhaps in another regular journal.

Additional comments:

Regarding the equations (3)-(5) in the Methods section, should "h" be replaced by \hbar ?

In the Methods section, why does equation (4) (interaction Hamiltonian) include an imaginary number i ?

In the response to referee #1's 1st comment, the intrinsic radiative lifetime of WS₂ is observed to be 250 ± 50 fs, while the lifetime of the Au metasurface is determined to be 1.8 ± 0.05 ps. The linewidth of the exciton resonance of WS₂ is narrower than that of the Au metasurface. Could the authors clarify why the intrinsic radiative lifetime of WS₂ is shorter?

In the response to referee #2's 4th comment, the manuscript states that the deviation between the spectral line splitting of SHG and dark-field scattering most likely arises from measurement errors in the strength of the SHG signal at the spectral edge of the photon detector. However, in general, the detector should not affect the value of spectral splitting. Could the authors provide further clarification on this discrepancy?

Fig. S15 (i.e., Fig. S19 in revised manuscript) is not cited in the main text. In addition, many supporting materials in the SI are also not cited in the main text.

In the figure caption of Fig. S19 in the revised version, normalized PL intensity is presented in both Fig. S19 a and Fig. S19 b. It is necessary for the author to explain the difference between the two to avoid confusion for the reader.

Response to Reviewers' Comments:

We would like to thank reviewer #2 for his/her careful reviewing of our work and for his/her valuable suggestions. Below, we provide a detailed point-by-point response that we believe addresses the reviewer's questions and comments. The reviewer's comments are in *italic*, and the authors' responses are in **Roman**.

Referee #2:

The authors have addressed the previous concerns and made notable improvements to the manuscript. While the observation of ultra-strong coupling between 2D material and plasmonic metasurface is impressive, it appears that the potential influence may not be as extensive as initially anticipated. The deterministic ultra-strong coupling between a single nanogap and 2D material remains unproven, as the observed Rabi splitting is the average value of an ensemble of detuned strong coupling systems. Additionally, while the wide linewidth does not impact the definition of ultra-strong coupling, it does limit the potential applications. Therefore, I believe this work could be published, but perhaps in another regular journal.

Response:

We would like to thank the reviewer for his/her efforts in reviewing our work.

- (1) Regarding the potential impact of USC between 2D materials and plasmonic metasurfaces, we first note that with USC, standard approximations break down and allow processes that do not preserve the number of excitations in the system, leading to a ground state containing virtual excitations (see Fig. 3d in the main text). The main reasons for the significant importance of USC include
 - a) More efficient light-matter interactions: USC enhances absorption, emission and nonlinear optical effects, enabling highly efficient photonic devices. USC enables the exploration of novel quantum phenomena and materials manipulation.
 - b) Better equipment performance: Upgrading from SC to USC results in better equipment performance, such as faster control and response, even with shorter component lifetimes.
 - c) Improved sensing and detection: Enhanced light-matter interaction improves the sensitivity and performance of sensors and detectors for applications in environmental monitoring, medical diagnostics and chemical sensing.
 - d) New quantum states: Ground-state energy modification enables applications in quantum information processing, sensing and communication, with tunable functionalities such as ultrafast switching and efficient energy transfer.

- e) Quantum simulation: USC systems serve as versatile platforms for the simulation of complex quantum phenomena, providing insights into fundamental dynamics and exploring new quantum phases.

In summary, USC between 2D materials and plasmonic metasurfaces provides a versatile platform and a frontier in quantum optics, offering opportunities to explore new optical phenomena, design advanced photonic devices and advance various technological applications in photonics and optoelectronics.

- (2) Regarding the unproven deterministic ultra-strong coupling between a single nanogap and a 2D material and the observed Rabi splitting as an average of an ensemble of detuned strong coupling systems, we emphasize:

- a) Resilience to fabrication variability: ensemble averaging over detuned strong coupling systems provides resilience to fabrication variability and imperfections. Achieving deterministic USC in each individual nanogap 2D material system may be challenging due to fabrication variations, but ensemble averaging accounts for these discrepancies.
- b) Statistical insights: ensemble averaging provides statistical insights into the collective behaviour of strongly coupled systems, identifying trends and guiding device optimisation.
- c) Device design flexibility: ensemble averaging accounts for variations in device parameters, facilitating rapid prototyping and new concept exploration.
- d) Scalability and efficiency: ensemble averaging streamlines manufacturing and reduces production costs by simplifying fabrication and quality control processes.
- e) Practicality: ensemble averaging is relevant to practical applications where robustness, scalability and cost-effectiveness are critical.

In summary, while deterministic ultra-strong coupling provides precise control, ensemble averaging provides valuable insight, flexibility and scalability benefits, particularly for practical device implementation and fabrication.

- (3) Regarding the comment on the impact of the linewidth on USC and its applications, we first emphasize that USC is characterized by light and matter excitations in the ground state, which depend more on the coupling strength than on the linewidth. In response to the point that coupling strength plays a more critical role than linewidth in the real applications of USC, we highlight the advantages of USC as follows.

- a) More efficient interactions: Due to the significant coupling strength, USC facilitates more efficient interactions compared to SC, which is crucial for

improving device performance, such as faster control and response in electro-optical devices.

- b) Improved device efficiency: Switching from SC to USC improves device efficiency at shorter component lifetimes.
- c) Emerging applications of USC: nonlinear optics, superconductivity, metamaterials and materials science.

In summary, USC, characterized by its reliance on coupling strength, offers enhanced device performance and access to quantum effects across multiple domains, highlighting its importance in advancing technological applications.

Comment #1:

Regarding the equations (3)-(5) in the Methods section, should "h" be replaced by \hbar ?

Response: We used \hbar in equations (3)-(5) in our original manuscript.

Comment #2:

In the Methods section, why does equation (4) (interaction Hamiltonian) include an imaginary number i ?

Response: Since we used equation $H_{int} = i\hbar g(\hat{a}^\dagger + \hat{a})(\hat{b} - \hat{b}^\dagger)$ to express the interaction Hamiltonian, we need to include ' i '. ' i ' is not needed for $H_{int} = \hbar g(\hat{a}^\dagger + \hat{a})(\hat{b} + \hat{b}^\dagger)$.

Note that the quantized electric field operator is given by $\hat{\mathbf{E}} = i \sum_k E_k (\mathbf{u}_k \hat{b} - \mathbf{u}_k^* \hat{b}^\dagger)$, where the complex pre-factor is sometimes absorbed in the mode function \mathbf{u} such that $\mathbf{f} = i\mathbf{u}$, and the quantized electric field operator becomes $\hat{\mathbf{E}} = \sum_k E_k (\mathbf{f}_k \hat{b} + \mathbf{f}_k^* \hat{b}^\dagger)$. The interaction of a two level quantum system with a single mode of the electric field is then given as $H_{int} = i\hbar g(\hat{a}^\dagger + \hat{a})(\hat{b} - \hat{b}^\dagger) = \hbar g'(\hat{a}^\dagger + \hat{a})(\hat{b} + \hat{b}^\dagger)$.

Comment #3:

In the response to referee #1's 1st comment, the intrinsic radiative lifetime of WS2 is observed to be 250 ± 50 fs, while the lifetime of the Au metasurface is determined to be 1.8 ± 0.05 ps. The linewidth of the exciton resonance of WS2 is narrower than that of the Au metasurface. Could the authors clarify why the intrinsic radiative lifetime of WS2 is shorter?

Response: We thank the referee for this constructive comment. The ultrafast dynamics of both uncoupled and coupled systems have been studied at room temperature using pump-probe transient absorption spectroscopy with pumping at 590 nm. The observed lifetime value (1.8 ± 0.05 ps) extracted from the differential transition ($\Delta T/T$) spectra of the bare Au metasurface originates mainly from the electron-phonon coupling with a partial contribution from the electron-electron scattering, leading lattice heating. We note here that the true plasmon lifetime of the Au metasurface, which is expected to be less than 10

fs, could not be captured in our experiments due to the time resolution limitations of our setup.

Comment #4:

In the response to referee #2's 4th comment, the manuscript states that the deviation between the spectral line splitting of SHG and dark-field scattering most likely arises from measurement errors in the strength of the SHG signal at the spectral edge of the photon detector. However, in general, the detector should not affect the value of spectral splitting. Could the authors provide further clarification on this discrepancy?

Response: Ideally, the detector should not affect the spectral split. However, practical issues such as sensitivity and calibration introduce uncertainties. Inaccuracies occur when the spectral edge of the detector fails to capture the full SHG signal, particularly at certain wavelengths, causing deviations in the spectral split between SHG and dark-field scattering. Essentially, the inefficiency of the SHG detector at the peak of the scattered spectrum results in a shift of the spectrum, leading to a mismatch between the detected SHG and scattered spectrum peaks.

Comment #5:

Fig. S15 (i.e., Fig. S19 in revised manuscript) is not cited in the main text. In addition, many supporting materials in the SI are also not cited in the main text. In the figure caption of Fig. S19 in the revised version, normalized PL intensity is presented in both Fig. S19 a and Fig. S19 b. It is necessary for the author to explain the difference between the two to avoid confusion for the reader.

Response: Thank you for bringing this to our attention. We have checked and ensured that all figures in the Supplementary Information (SI) are appropriately cited in the revised manuscript.

To address potential confusion, we have updated the caption of Fig. S19 (b) (Fig. S15 (b) in revised SI) from "Normalized PL intensity" to "Normalized PLE" to clarify its content. We provide both normalized PL spectra and PLE spectra to account for the absence of the upper polariton branch in the PL spectra of strongly coupled systems. This is because in the strongly coupled systems, part of the detected PL ($PL_{WS_2 \text{ on metasurface}}$) is emitted by the radiation of plasmon-exciton hybrid modes and part of the detected PL signal is due to the background PL from the uncoupled excitons ($PL_{WS_2 \text{ on polymer}}$). This complicates the extraction of strong coupling information from the PL spectra. By separating PL emission attributed to plasmon-exciton hybrid modes from that of uncoupled excitons (i.e., $PLE = (PL_{WS_2 \text{ on metasurface}} - PL_{WS_2 \text{ on polymer}}) / PL_{WS_2 \text{ on polymer}}$), we can reveal the strong coupling induced spectral features of PL. Thus, both PL and PLE spectra are included.